# Thermally boosted upconversion and downshifting luminescence in Sc$_2$(MoO$_4$)$_3$:Yb/Er with two-dimensional negative thermal expansion

Jinsheng Liao [1,5✉], Minghua Wang[1,5], Fulin Lin[2,3,5], Zhuo Han[1,5], Biao Fu[1], Datao Tu [2✉], Xueyuan Chen [2✉], Bao Qiu [4✉] & He-Rui Wen[1]

Rare earth (RE$^{3+}$)-doped phosphors generally suffer from thermal quenching, in which their photoluminescence (PL) intensities decrease at high temperatures. Herein, we report a class of unique two-dimensional negative-thermal-expansion phosphor of Sc$_2$(MoO$_4$)$_3$:Yb/Er. By virtue of the reduced distances between sensitizers and emitters as well as confined energy migration with increasing the temperature, a 45-fold enhancement of green upconversion (UC) luminescence and a 450-fold enhancement of near-infrared downshifting (DS) luminescence of Er$^{3+}$ are achieved upon raising the temperature from 298 to 773 K. The thermally boosted UC and DS luminescence mechanism is systematically investigated through in situ temperature-dependent Raman spectroscopy, synchrotron X-ray diffraction and PL dynamics. Moreover, the luminescence lifetime of $^4$I$_{13/2}$ of Er$^{3+}$ in Sc$_2$(MoO$_4$)$_3$:Yb/Er displays a strong temperature dependence, enabling luminescence thermometry with the highest relative sensitivity of 12.3%/K at 298 K and low temperature uncertainty of 0.11 K at 623 K. These findings may gain a vital insight into the design of negative-thermal-expansion RE$^{3+}$-doped phosphors for versatile applications.

[1] School of Chemistry and Chemical Engineering/Jiangxi Provincial Key Laboratory of Functional Molecular Materials Chemistry, Jiangxi University of Science and Technology, Ganzhou, Jiangxi 341000, P. R. China. [2] CAS Key Laboratory of Design and Assembly of Functional Nanostructures, and Fujian Key Laboratory of Nanomaterials, Fujian Institute of Research on the Structure of Matter, Chinese Academy of Sciences, Fuzhou, Fujian 350002, China. [3] Xiamen Institute of Rare Earth Materials, Haixi Institute, Chinese Academy of Sciences, Xiamen 361021, China. [4] Ningbo Institute of Materials Technology & Engineering (NIMTE), Chinese Academy of Sciences, Ningbo, Zhejiang 315201, P. R. China. [5] These authors contributed equally: Jinsheng Liao, Minghua Wang, Fulin Lin, Zhuo Han. ✉email: jsliao1209@126.com; dttu@fjirsm.ac.cn; xchen@fjirsm.ac.cn; qiubao@nimte.ac.cn

The photoluminescence (PL) intensity of RE$^{3+}$-doped phosphors is usually quenched with increasing temperature, which is referred to as positive thermal quenching[1–4]. This phenomenon has been frequently observed in most phosphors due to the vibration aggravation with temperature to promote the domination of nonradiative multiphonon transition probability (Fig. 1a). Such a phenomenon greatly limits the application of luminescent materials in the high-temperature region, which causes degeneration of device performance and eventually system failure. If no obvious decrease of PL intensity occurs on heating, the phosphor would show zero-thermal-quenching performance[5,6]. This pathway can occur in the compensation of emission loss through the polymorphism modification of the host, wherein the energy transfer (ET) from electron-hole pairs to the excited state energy levels results in radiative recombination in close proximity at the thermally activated defect levels (Fig. 1b).

Hitherto, several phosphors have been reported to exhibit abnormal thermo-enhanced luminescence during heating[7–10]. Most of such abnormal thermally enhanced UC luminescence is observed in RE$^{3+}$-doped inorganic materials with three-dimensional negative-thermal-expansion (NTE) characteristics[9], where all the three cell parameters of the doped crystals shrink at

elevated temperature. Such shrinkage may induce the decrease of distance between the sensitized ions and the activated ions to improve the ET efficiency, resulting in the enhancement of luminescence intensity[9–11]. Meanwhile, such a three-dimensional compression may also promote the dissipation of the excitation energy in all directions of crystal sublattice to the lattice/surface defects, which deteriorates the luminescent emission of RE$^{3+}$ ions. At present, only a few NTE host materials are demonstrated to be suitable for RE$^{3+}$ doping and their luminescence performance is usually too poor to fulfill their practical applications[12–16]. As such, more and more works have been devoted to the exploration of novel inorganic materials with thermo-enhanced luminescence properties (Fig. 1c)[17–22].

In this work, we report the synthesis and characterization of a phosphor based on Yb$^{3+}$/Er$^{3+}$ co-doped NTE matrix of Sc$_2$(MoO$_4$)$_3$ with a unique two-dimensional NTE coefficient ($\alpha_a = -8.62 \times 10^{-6}$/K, $\alpha_b = 4.25 \times 10^{-6}$/K, $\alpha_c = -6.35 \times 10^{-6}$/K) (Fig. 1d). The interatomic distance of RE$^{3+}$ ions in the matrix can be manipulated with temperature to realize the thermal enhanced both UC and DS emission of Er$^{3+}$ upon 980-nm excitation. We apply in situ temperature-dependent Raman spectroscopy, synchrotron X-ray diffraction, and luminescence dynamics to reveal

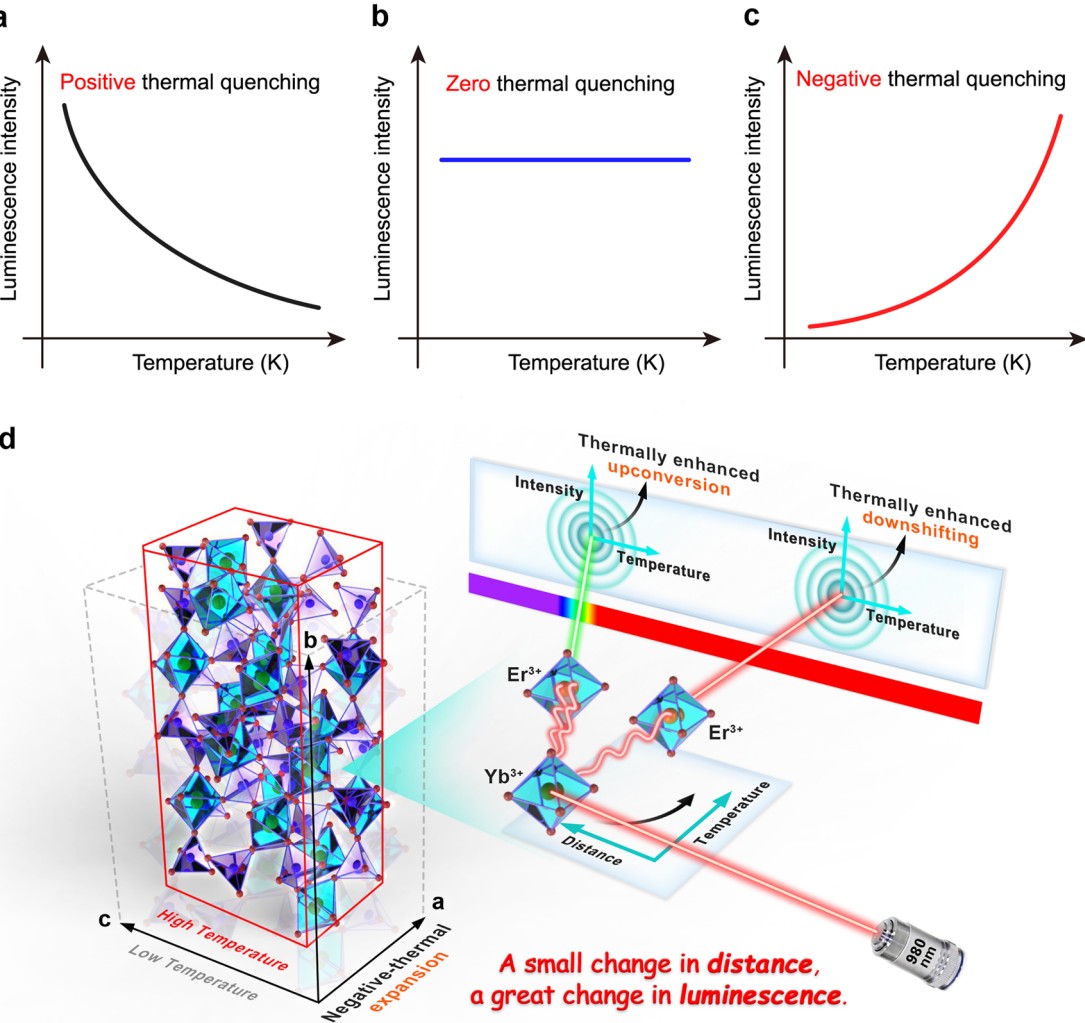

**Fig. 1 Scheme of thermal-dependence effects in phosphor. a** Positive thermal quenching phenomenon. **b** Zero thermal quenching phenomenon. **c** Negative thermal quenching (thermal enhanced) phenomenon. **d** Schematic illustration of the negative-thermal-index in Sc$_2$(MoO$_4$)$_3$:Yb/Er, indicating that both the *a*-axis and *c*-axis contract while the *b*-axis expands at high temperature. Upon 980 nm laser excitation, simultaneously thermally enhanced upconversion and downshifting luminescence of Er$^{3+}$ are achieved in Yb$^{3+}$/Er$^{3+}$ co-doped Sc$_2$(MoO$_4$)$_3$ with a two-dimensional negative-thermal expansion.

the mechanism of negative-thermal effects. We demonstrate that the thermally boosted UC and DS emissions can be achieved benefitting from the lattice shrinkage and confined energy migration at elevated temperatures. Moreover, the NIR-II PL lifetime of $Er^{3+}$ in $Sc_2(MoO_4)_3$:Yb/Er phosphors is found to be increased by more than two orders of magnitude as the temperature increases from 298 to 623 K. As such, we employ the proposed phosphors for luminescent lifetime-based temperature sensing.

## Results

**Negative-thermal expansion in $Sc_2(MoO_4)_3$:Yb/Er**. $Sc_2(MoO_4)_3$:Yb/Er phosphors were synthesized via a sol-gel method. The basic structures of the $Sc_2(MoO_4)_3$:Yb/Er samples with different substituted concentrations were detected by X-ray diffraction (XRD) patterns (Supplementary Fig. 1). All the structures of the as-prepared samples are well consistent with the orthorhombic $Sc_2(MoO_4)_3$ (ICSD#20838) without any observable impurities. Their chemical compositions and morphologies were determined by energy-dispersive X-ray spectroscopy (EDS) and scanning electron microscopy (SEM), respectively (Supplementary Fig. 2). It can be observed that the as-prepared $Sc_2(MoO_4)_3$:Yb/Er samples are microcrystals with sizes of 1–2 μm and Sc, Yb, Er, Mo and O elements are homogeneously distributed in the matrix.

Figure 2a, b show the two-dimensional topographical mappings of temperature-dependent in situ SXRD patterns for (020), (200), and (002) peaks within the temperature range from 298 to 673 K. When the temperature increases, the (020) diffraction peak shifts to lower angle, indicating the thermal expansion in $b$-axis. A continuous redshift of the (200) diffraction peak presents a thermal contraction in the $a$-axis. For $c$-axis that correspondings to the (002) diffraction peak, it exhibits positive-thermal expansion (PTE) first and then NTE with the temperature higher than 353 K. Such a transformation from PTE to NTE can be attributed to the existence of water molecules in the $Sc_2(MoO_4)_3$:20%Yb/1%Er phosphors as revealed by thermogravimetry (TG) analysis and in situ temperature-dependent Fourier transform infrared (FTIR) spectroscopy (Supplementary Fig. 3). When the temperature increases up to 353 K, water molecules begin to remove from the microchannels to recover the NTE phenomena[23]. The lattice parameters were refined and calculated based on the in situ temperature-dependent SXRD patterns (Fig. 2c and Supplementary Fig. 4). Accordingly, their expansion coefficients were calculated to be $\alpha_a = -8.62 \times 10^{-6}$/K, $\alpha_b = 4.25 \times 10^{-6}$/K, $\alpha_c = -6.35 \times 10^{-6}$/K. Note that almost all the reported $RE^{3+}$-doped NTE phosphors show a three-dimensional negative-thermal expansion, while $Sc_2(MoO_4)_3$:Yb/Er phosphors exhibit a unique two-dimensional negative-thermal expansion in a wide temperature range.

Figure 2d shows the distances of $RE^{3+}$-$RE^{3+}$ (RE = Sc/Yb/Er) ions at different axes estimated by the crystallographic information file (CIF) of *Rietveld* refinement of the SXRD patterns (Supplementary Fig. 4). With the temperature higher than 353 K, the distance of $RE^{3+}$-$RE^{3+}$ ions steadily shortened along $a$ and $c$ axes and continuously lengthened along $b$ axis. Under such circumstances, the migration of excitation energy between $RE^{3+}$-$RE^{3+}$ ions will be gradually minimized along $b$ axis. Meanwhile, the contracted distances of $RE^{3+}$-$RE^{3+}$ ions along $a/c$-axis at elevated temperature may facilitate the ET efficiency between $Yb^{3+}$ and $Er^{3+}$ [24]. Thus, more energy migration between $RE^{3+}$-$RE^{3+}$ ions is confined to two dimensions on the (020) lattice plane for producing intense PL emissions at elevated temperature[25].

To shed more light on the NTE behavior of $Sc_2(MoO_4)_3$:Yb/Er phosphors, a rigid unit mode model is illustrated for $RE_2Mo_3O_{12}$ (Fig. 2e)[16,26]. With the temperature from 420 to 620 K, the distortion of the $REO_6$ octahedron and twist of the RE-O-Mo ratio in the distance of RE-Mo along $a$ and $c$ axes decreased by 0.13% and 0.15%, while the ratio in the distance of RE-Mo along $b$ axis increased by 0.02% (Fig. 2f), which resulted in the shrinkage of lattice and distortion of local symmetry of $RE^{3+}$ with increasing the temperature. The angle of RE-Mo-RE nearly keeps unchanged at the temperature range of 420–620 K (Fig. 2g), which indicates that the structure of $Sc_2Mo_3O_{12}$ is rigid.

Raman spectroscopy is a valuable tool to study the phonon modes of NTE materials and evaluate the change of local structure[23,27,28]. In situ temperature-dependent Raman spectra are shown in Fig. 2h. The Raman peaks of 330, 836, and 945 $cm^{-1}$ are characteristic of the hydrated orthorhombic structure, indicative of water species residing in the microchannels of $Sc_2(MoO_4)_3$:20%$Yb^{3+}$/1%$Er^{3+}$ [23]. These peaks remain nearly unchanged below 348 K, which reveals that the water molecules were not removed from the microchannels. As the temperature increases from 348 to 423 K, these peaks become weaker. In addition, the peak at 945 $cm^{-1}$ vanishes above 423 K, which demonstrates that water molecules were completely removed. These results are in agreement with the obtained results of the TG, FTIR spectra, as well as in situ temperature-dependent SXRD patterns (Supplementary Figs. 3 and 4). The Raman peak with a frequency of 341 $cm^{-1}$ ($v_4$) exhibits a blue shift as the temperature increases (Fig. 2h), suggesting that this mode is the origin of NTE in $Sc_2(MoO_4)_3$:Yb/Er [29,30]. The median frequency at 510 $cm^{-1}$ and 579 $cm^{-1}$ was induced by the disorder of $MoO_4$ tetrahedra because of the incorporation of the $Yb^{3+}$ and $Er^{3+}$ into the lattice[31,32], which would affect the local structure of the $Yb^{3+}$ and $Er^{3+}$ ions. Note that the temperature coefficients of high phonon frequencies (511, 579, 813, 979 $cm^{-1}$) are negative above 348 K (Fig. 2i, j), verifying strong anharmonic stretching/bending of $MoO_4$ tetrahedra[33]. These results suggest that high phonon frequencies also contribute to NTE of the phosphors[23]. Moreover, the reduction of maximum phonon energy (979 $cm^{-1}$) may benefit the luminescence of $RE^{3+}$ at elevated temperatures due to the suppressed nonradiative relaxation.

**Thermally enhanced upconversion emission**. Figure 3a shows the UC emission spectra of the $Sc_2(MoO_4)_3$:20%Yb/1%Er phosphors upon 980-nm laser excitation at different temperatures. A weak red emission band peaking at 654 nm is attributed to the $^4F_{9/2} \to {}^4I_{15/2}$ transition of $Er^{3+}$ [24]. With the increase in temperature from 298 to 773 K, the red emission of $Er^{3+}$ was boosted by 14-fold, due to the enhanced back energy transfer from $Er^{3+}$ to $Yb^{3+}$ with shortened $Yb^{3+}$-$Er^{3+}$ interatomic distance[34]. Besides, two intense green emission bands peaking at 522 and 558 nm are observed (Fig. 3b), which correspond to the $^2H_{11/2} \to {}^4I_{15/2}$ and $^4S_{3/2} \to {}^4I_{15/2}$ transitions of $Er^{3+}$, respectively. In Fig. 3c, two temperature regions for the UC emissions can be observed. When the temperature increases from 298 to 473 K, the overall green emission intensity increases by 26-fold. When the temperature increases from 473 to 773 K, the green UC intensity can be further enhanced. Overall, the green UC intensity increases by 45-fold from 298 to 773 K. To explicitly show the changes of brightness, we provide the photographs of logo for Jiangxi University of Science and Technology (JXUST) made by $Sc_2(MoO_4)_3$:20%Yb/1%Er phosphors with increasing temperature (Fig. 3d). Upon 980-nm laser diode excitation, it can be seen that the logo becomes brighter with elevating the temperature.

**Thermally enhanced downshifting emission**. To investigate the NTE effect on the DS emission, the temperature-dependent DS excitation spectra of the $Sc_2(MoO_4)_3$:20%Yb/1%Er phosphors were measured by monitoring the emission of $Er^{3+}$ at 1538 nm (Supplementary Fig. 5). Accordingly, the intensity of the excitation peak increased with the temperature rising from 298 to

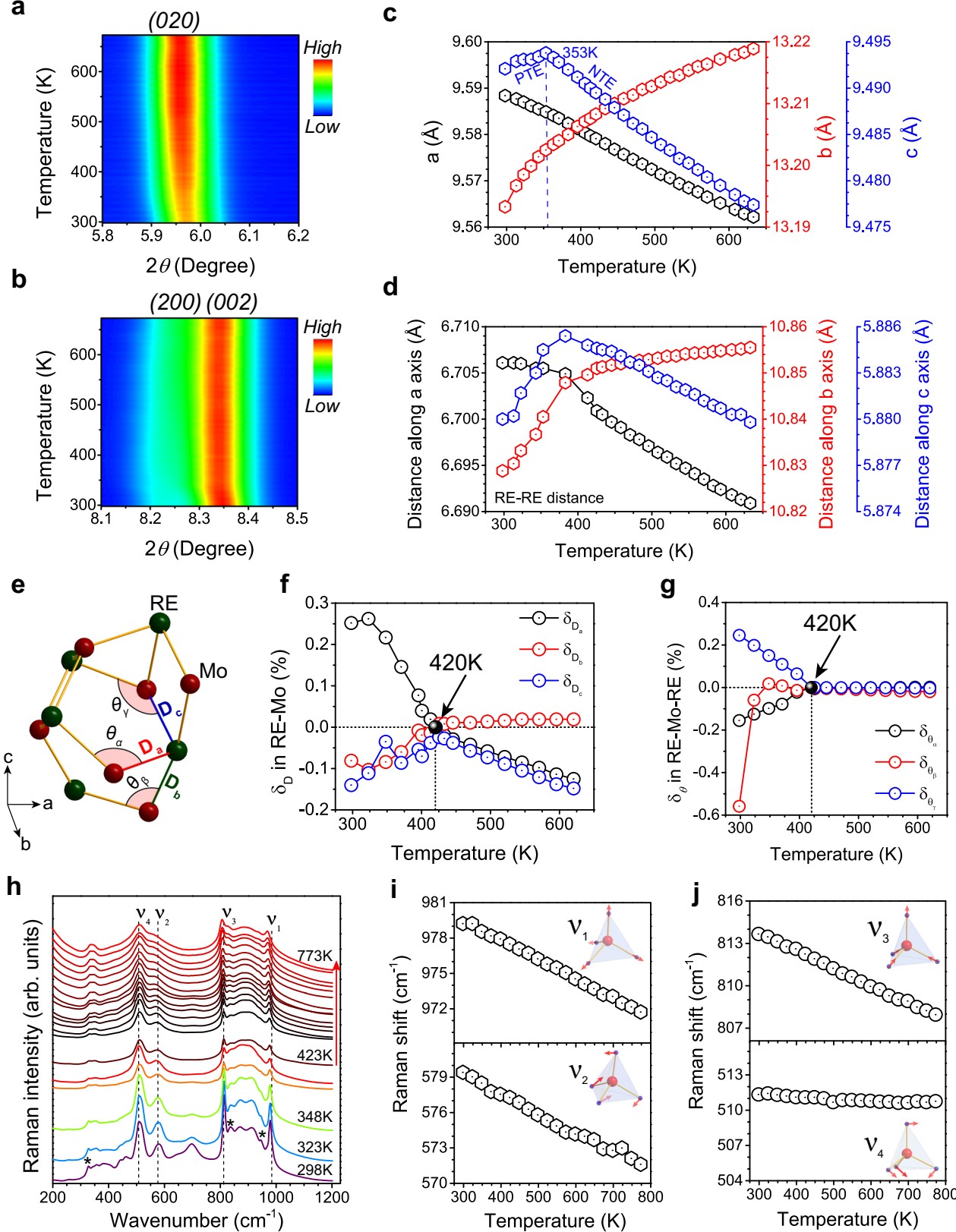

573 K, which may benefit the luminescence emission of $Er^{3+}$ at a higher temperature.

Figure 4a shows the temperature-dependent DS emission spectra of the $Sc_2(MoO_4)_3$:20%Yb/1%Er phosphors upon excitation at 980 nm. The spectra exhibit characteristic near-infrared II (NIR-II) emission of $Er^{3+}$ peaking at 1538 nm, which originates from the $^4I_{13/2} \rightarrow ^4I_{15/2}$ transition. The emission intensity also displays a significant enhancement with increasing the temperature from 298 to 773 K (Fig. 4b), where a 450-fold enhancement of the integrated luminescence intensity was achieved (Fig. 4c). Figure 4d shows the temperature-dependent NIR photographs of logo for JXUST made by $Sc_2(MoO_4)_3$:20%Yb/1%Er phosphors.

**Fig. 2 Negative-thermal expansion in Sc₂(MoO₄)₃:20%Yb/1%Er. a, b** Two-dimensional topographical mapping of temperature-dependent in situ synchrotron X-ray diffraction patterns of (020), (200), and (002) peaks with the temperature from 298 K to 673 K. **c** Temperature-dependent unit cell parameters $a$, $b$, and $c$ of the Sc₂(MoO₄)₃:20%Yb/1%Er phosphors derived from in situ SXRD patterns. **d** Temperature-dependent proximate distances of RE-RE (RE=Sc/Yb/Er) along different axes derived from the unit cell structure. **e** Rigid unit mode model for Sc₂Mo₃O₁₂:20%Yb³⁺/1%Er³⁺ extracted from the unit cell structure. **f** Temperature-dependent ratio $\delta D$ of RE-Mo distances marked in the model ($\delta D = \frac{D_T - D_{420}}{D_{420}} \times 100\%$, $D_T$ and $D_{420}$ stand for RE-Mo distances of the given temperature and 420 K, respectively). **g** Temperature-dependent ratio $\delta\theta$ of RE-Mo-RE angles marked in the model ($\delta\theta = \frac{\theta_T - \theta_{420}}{\theta_{420}} \times 100\%$, $\theta_T$ and $\theta_{420}$ stand for RE-Mo-RE angles of the given temperature and 420 K, respectively). **h** Temperature-dependent in situ Raman spectra with the temperature from 298 to 773 K. * stands for the characteristic Raman peak of the hydrated orthorhombic structure. **i** and **j**. Raman shifts at different temperatures. $\nu_1$, $\nu_2$, $\nu_3$, and $\nu_4$ stand for symmetric stretching, symmetric bending, asymmetric stretching, and asymmetric bending mode of MoO₄ tetrahedra, respectively.

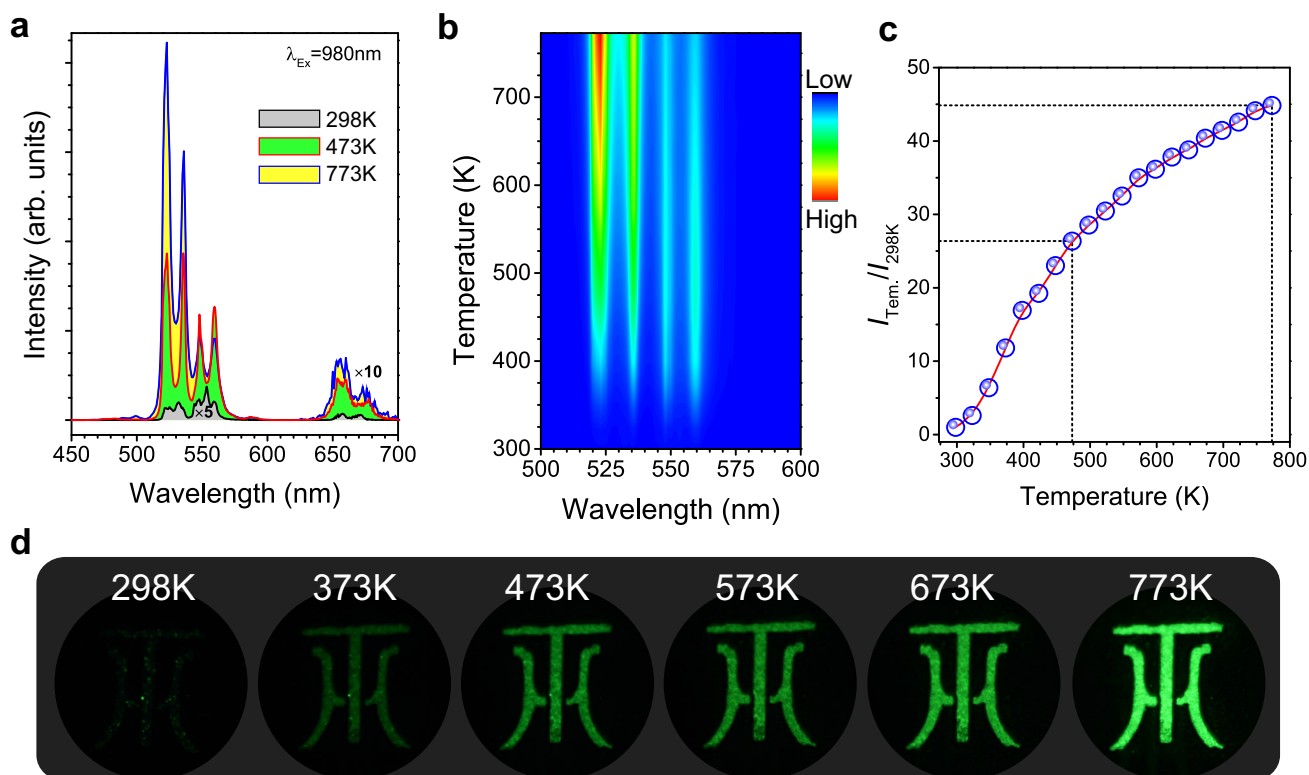

**Fig. 3 Thermally enhanced upconversion emission. a** UC spectra of Sc₂(MoO₄)₃:20%Yb/1%Er phosphor with different temperatures. **b** Two-dimensional UC emission topographical mapping within the temperature from 298 to 773 K. **c** Normalized UC emission intensity as a function of temperature. **d** UC photographs of Sc₂(MoO₄)₃:20%Yb/1%Er phosphors at various temperatures recorded by the Nikon D750 camera upon 980 nm excitation.

Upon 980-nm laser excitation, the logo of JXUST also became brighter by a NIR camera with the increase of temperature. Due to the low pixel size of current near-infrared cameras, the shortened exposure time may reduce light saturation. As the temperature increases, the DS luminescence sharply increases. Therefore, we chose different exposure times in different temperature regions to take the photographs.

As we know, the famous orthorhombic-phase YF₃:20%Yb/1%Er phosphor with the same crystallographic system of Sc₂(MoO₄)₃ is considered to be one of the most efficient UC and DS emitting phosphors[35]. To compare the emission intensity of YF₃:20%Yb/1%Er and Sc₂(MoO₄)₃:20%Yb/1%Er phosphors, the temperature-dependent UC/DS spectra of them were measured under otherwise identical conditions (Supplementary Figs. 6 and 7). It can be observed that the UC/DS intensity of YF₃:20%Yb/1%Er phosphor decreased continuously with increasing the temperature, while the UC/DS intensity of Sc₂(MoO₄)₃:20%Yb/1%Er phosphor increased markedly with elevating the temperature. Specifically, the overall UC/DS luminescence intensity of YF₃:20%Yb/1%Er is much higher than that of Sc₂(MoO₄)₃:20%Yb/1%Er at 298 K. Nevertheless, the

integrated UC and DS luminescence intensity of Sc₂(MoO₄)₃:20% Yb/1%Er phosphor are 4.5 and 12.9 times higher than that of YF₃:20%Yb/1%Er counterpart at 773 K, respectively. These results explicitly validate the superiority of the Sc₂(MoO₄)₃:Yb/Er as luminescent materials over existing PTE phosphors, particularly at high temperatures.

**Thermally enhanced photoluminescence mechanism.** To shed more light on the mechanism responsible for the thermal-enhanced UC and DS emissions of Sc₂(MoO₄)₃:20%Yb/1%Er with two-dimensional NTE, the temperature-dependent luminescence of Yb³⁺ was investigated. Figure 5a displays the excited state (²F₅/₂) lifetime of Yb³⁺ in Sc₂(MoO₄)₃:20%Yb/1%Er. It exhibits a marked increase in the PL lifetime from 16.62 to 278 μs as the temperature is raised from 298 to 473 K. Generally, the radiation trapping of Yb³⁺ may lengthen the PL lifetime in phosphors, because the lattice contraction would shorten the distance of Yb³⁺-Yb³⁺ at elevated temperature to promote the radiation trapping of Yb³⁺ (Fig. 2d)[36,37]. In the Yb³⁺-Er³⁺ co-

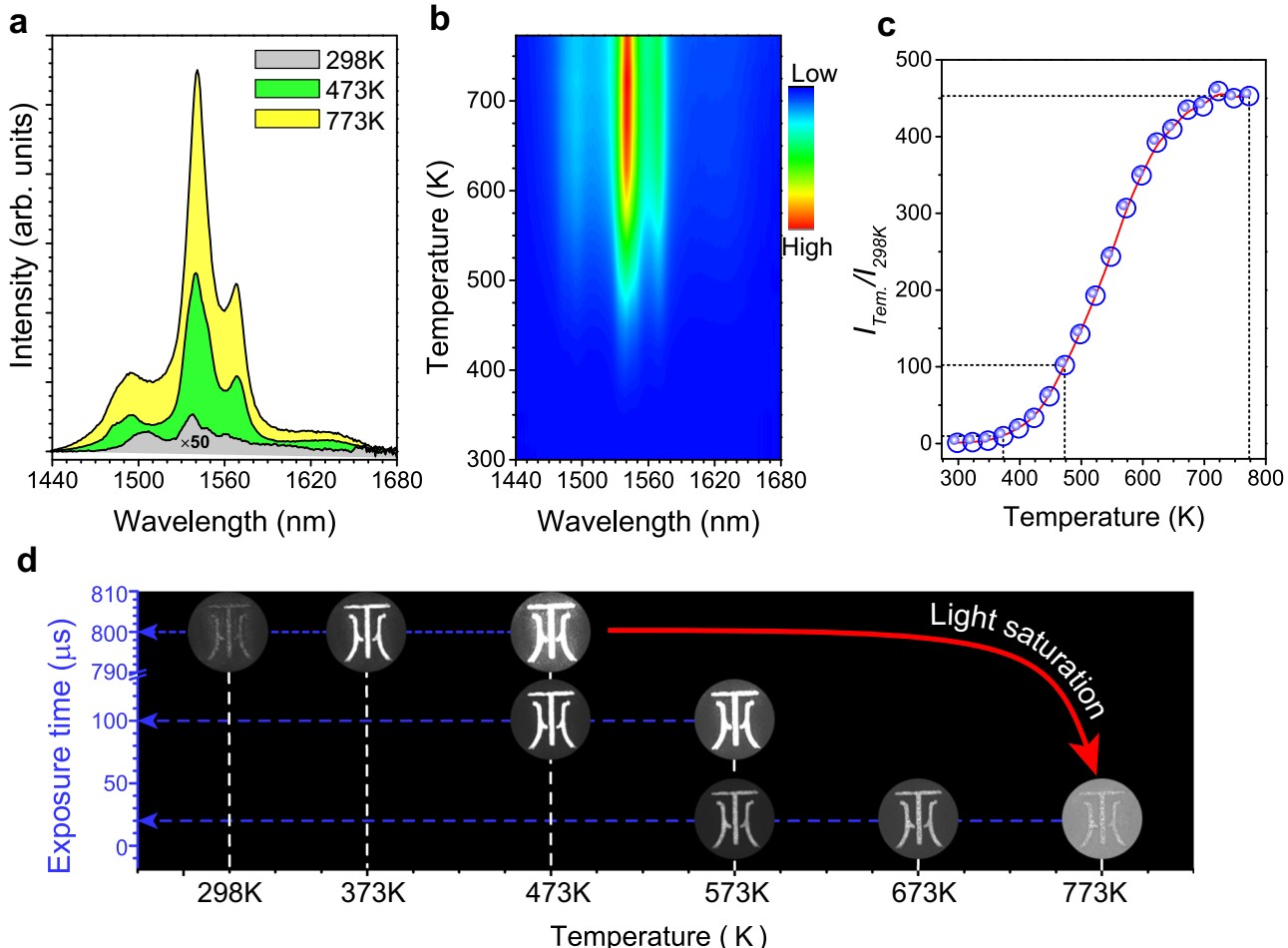

**Fig. 4 Thermally enhanced downshifting emission. a** DS emission spectra of $Sc_2(MoO_4)_3$:20%Yb/1%Er phosphor with different temperatures. **b** Two-dimensional DS emission topographical mapping with the temperature from 298 to 773 K. **c** Normalized DS emission intensity as a function of temperature. **d** Photographs of $Sc_2(MoO_4)_3$:20%Yb/1%Er at various temperatures by a NIR camera. To avoid interference of 980 nm laser, a 1250 nm filter was added in the front of the Photonic Science InGaAs camera. The red arrow represents decreasing exposure times to avoid light saturation.

doped phosphors, $Yb^{3+}$ acts not only as radiation-trapping to store energy but also as a sensitizer to transfer the energy to $Er^{3+}$. These radiation trappings may enhance the emission of $Er^{3+}$. Moreover, the distance of $Yb^{3+}$-$Er^{3+}$ becomes shorter with the increase of temperature. The ET processes between sensitizer ($Yb^{3+}$) to activator ($Er^{3+}$) are usually considered to occur through dipolar-dipolar interactions, whose ET efficiency is proportional to $r^{-6}$ (r is the donor-acceptor distance)[38]. As such, the ET efficiency can be greatly improved at higher temperatures. Correspondingly, both the UC and DS emissions of $Sc_2(MoO_4)_3$:Yb/Er were significantly enhanced when the temperature increased.

Moreover, $Yb_2WO_6$:1%Er with normal PTE was adopted as a comparative investigation[9]. As expected, thermally quenched PL emission was detected when the temperature increased from 298 to 573 K (Supplementary Figs. 8 and 9), since the increased distance between $Yb^{3+}$ to $Er^{3+}$ would reduce the ET efficiency from $Yb^{3+}$ to $Er^{3+}$. To reveal the role of the sensitizer of $Yb^{3+}$, $Er^{3+}$ single-doped $Sc_2(MoO_4)_3$, and $Yb^{3+}$ single-doped $Sc_2(MoO_4)_3$ phosphors were also synthesized. From the temperature-dependent PL emission spectra of the $Sc_2(MoO_4)_3$:Er (5%) phosphors (Supplementary Fig. 10), the UC emission intensity of $Er^{3+}$ decreased with the increase of temperature upon 980-nm excitation without the ET from $Yb^{3+}$ to $Er^{3+}$ due to the detrimental cross-relaxation process between $Er^{3+}$ ions. Meanwhile, the NIR-II DS emission of $Er^{3+}$ for

$Sc_2(MoO_4)_3$:Er (5%) phosphors was improved by only 5.9-fold with the increase of temperature (Supplementary Fig. 11), which is greatly less than the enhanced factor of $Sc_2(MoO_4)_3$:20%Yb/1%Er phosphors. These results verify that the contribution of ET from the sensitizer ($Yb^{3+}$) to the activator ($Er^{3+}$) is essential for achieving efficient UC and DS luminescence of $Er^{3+}$.

For $Yb^{3+}$-doped $Sc_2(MoO_4)_3$ phosphors, the excited state ($^2F_{5/2}$) lifetime of $Yb^{3+}$ exhibits a similar change trend as that of $Sc_2(MoO_4)_3$:20%Yb/1%Er (Fig. 5a). Without the ET from $Yb^{3+}$ to $Er^{3+}$, the excited state ($^2F_{5/2}$) lifetime of $Yb^{3+}$ in $Sc_2(MoO_4)_3$:Yb phosphors was determined to be longer than that of $Sc_2(MoO_4)_3$:20%Yb/1%Er at the same temperature. The ET efficiency ($\eta_{ET}$) can be calculated from the following expression:[39]

$$\eta_{ET} = 1 - \frac{\tau_{Yb-Er}}{\tau_{Yb}} \qquad (1)$$

where $\tau_{Yb-Er}$ and $\tau_{Yb}$ are the PL lifetimes of $Yb^{3+}$ at 1051 nm in $Sc_2(MoO_4)_3$:Yb/Er and $Sc_2(MoO_4)_3$:Yb phosphors (Supplementary Fig. 12), respectively. Accordingly, it was determined that the ET efficiency from $Yb^{3+}$ to $Er^{3+}$ in $Sc_2(MoO_4)_3$:20%Yb/1%Er increased gradually when the temperature was raised from 298 to 473 K. For the temperature above 473 K, the ET efficiency kept almost constant (~55%). However, the overall UC and DS emission intensities of $Sc_2(MoO_4)_3$:Yb/Er continued to increase in this temperature region.

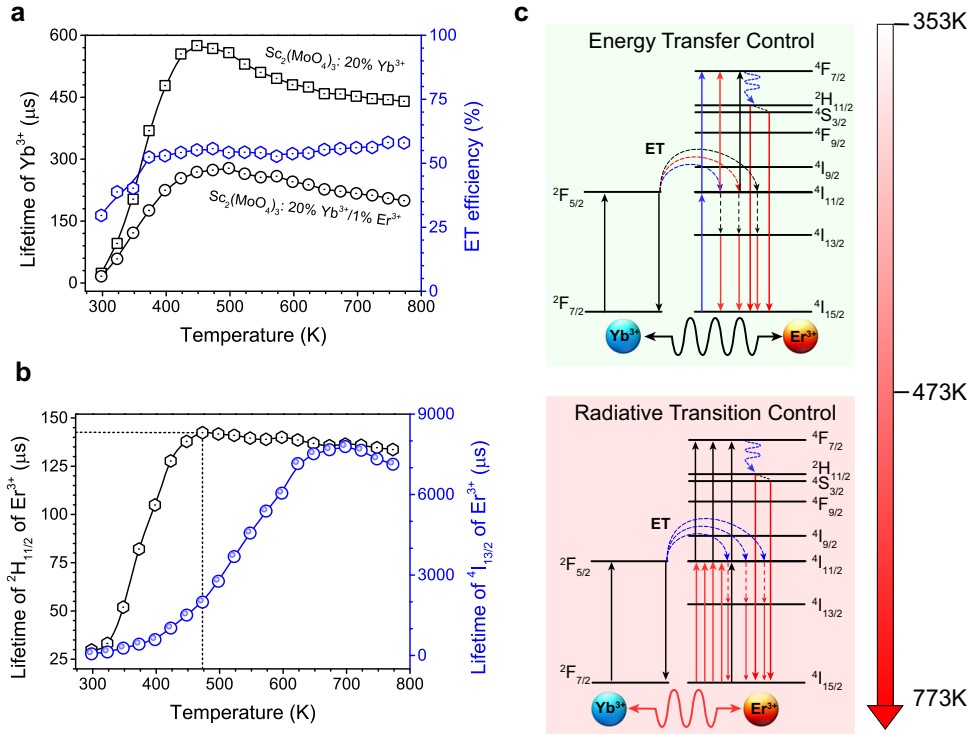

**Fig. 5 Thermally enhanced photoluminescence mechanism. a** Temperature-dependent luminescence lifetime of $^2F_{5/2}$ excited state of $Yb^{3+}$ in $Yb^{3+}/Er^{3+}$-co-doped and $Yb^{3+}$-doped $Sc_2(MoO_4)_3$, respectively. Temperature-dependent energy transfer efficiency of $Yb^{3+}$-to-$Er^{3+}$. **b** Temperature-dependent lifetimes of $^2H_{11/2}$ (522 nm) and $^4I_{13/2}$ (1538 nm) excited states of $Er^{3+}$ in $Sc_2(MoO_4)_3$:20%Yb/1%Er, respectively. **c** Energy level diagram of green UC and NIR DS emission showing the proposed temperature dependence of electronic transition and energy-transfer processes in $Sc_2(MoO_4)_3$:Yb/Er with two-dimensional negative thermal expansion.

To figure out this contradiction, the temperature-dependent UC PL lifetime of $^2H_{11/2}$ for $Er^{3+}$ was measured (Fig. 5b and Supplementary Fig. 13). It exhibited an increase from 29.77 to 51.99 μs with the temperature from 298 to 348 K, which implies that the nonradiative transition of $^2H_{11/2} \to ^4F_{9/2}$ was inhibited by alleviating the quenching effect of $OH^-$ center. It exhibited an obvious increase from 51.99 to 142.36 μs with the temperature from 348 to 473 K, which verifies that the nonradiative transition of $^2H_{11/2} \to ^4F_{9/2}$ was inhibited at elevated temperature. Nevertheless, only a tiny change was observed for the PL lifetime of $^2H_{11/2}$ for $Er^{3+}$ with the temperature above 473 K. Note that a similar trend was also observed for the PL lifetime of $Yb^{3+}$ in $Sc_2(MoO_4)_3$:Yb. The luminescence lifetime ($\tau$) of an excited state is determined by $\tau = 1/(A_r + A_{nr})$[40,41], where $A_r$ and $A_{nr}$ are the radiative and nonradiative transition probabilities of this energy level, respectively. Accordingly, it can be deduced that the increase of $A_r$ is nearly equal to the decrease of $A_{nr}$. The promoted $A_r$ and inhibited $A_{nr}$ of the $RE^{3+}$ dopants indicate that lattice distortion enhances a crystal field with odd parity and modifies the local symmetry of activator ions via the NTE effect[40]. Based on the above analysis, the thermally enhanced UC emission with the temperature from 298 to 473 K is mainly governed by the ET from $Yb^{3+}$ to $Er^{3+}$. For the temperature above 473 K, the thermally enhanced UC emission is mainly controlled by the radiation transition of $Er^{3+}$ that is strengthened with an increase of temperature (Fig. 5c).

To reveal the mechanism responsible for the negative-thermal DS luminescence of $Sc_2(MoO_4)_3$:Yb/Er, we measured the temperature-dependent PL decays of $^4I_{13/2}$ of $Er^{3+}$, which can be well fitted by a single-exponential function (Supplementary Fig. 13). When the temperature increased from 298 to 473 K, it was determined that the PL lifetime of $^4I_{13/2}$ level of $Er^{3+}$

increased markedly from 61.2 μs to 2004 μs. Meanwhile, the $Yb^{3+}$-$Er^{3+}$ ET efficiency of the $Sc_2(MoO_4)_3$:Yb/Er increased from 29% to 55% with increasing the temperature (Fig. 5a). According to the previous analysis of UC process, the thermally enhanced DS luminescence for this temperature region was mainly governed by the ET from $Yb^{3+}$ to $Er^{3+}$. Besides, the PL lifetime of $^4I_{13/2}$ level of $Er^{3+}$ increased from 2004 to 7789 μs as the temperature raised from 473 to 698 K. At higher temperature above 698 K, the PL lifetime of $^4I_{13/2}$ level of $Er^{3+}$ decreased, which indicated that the increase of $A_r$ was larger than the decrease of $A_{nr}$. As such, the NIR DS emission with the temperature from 473 to 773 K was also enhanced via the control of radiative transition (Fig. 5c).

**Photoluminescence lifetime-based thermal sensing.** The tunable PL lifetime of $Er^{3+}$ in $Sc_2(MoO_4)_3$:Yb/Er phosphors, which spans two orders of magnitude with the temperature from 298 to 623 K, offers a great opportunity for their application in lifetime-based luminescent thermometry (LLT). Previously, $Sc_2(MoO_4)_3$:Yb/Ho with the same host was used as a ratiometric thermometer based on the UC red-to-green emission intensity ratio[10]. Note that LLT is of special importance owing to several attractive merits over conventional intensity-based thermometry[42–46]. For instance, PL lifetime is insensitivity to the variation in phosphors concentration and excitation intensity, which enables LLT to circumvent the measurement error that is intrinsic and inevitable for intensity-based thermometry. In addition, for the conventional intensity-based thermometry, the intensity signal may become unreliable at different depths since the signal-to-noise ratio may be very low when the depth-induced quenching of the intensity signal is serious. By contrast, the lifetime signal remains essentially unaltered at different depths.

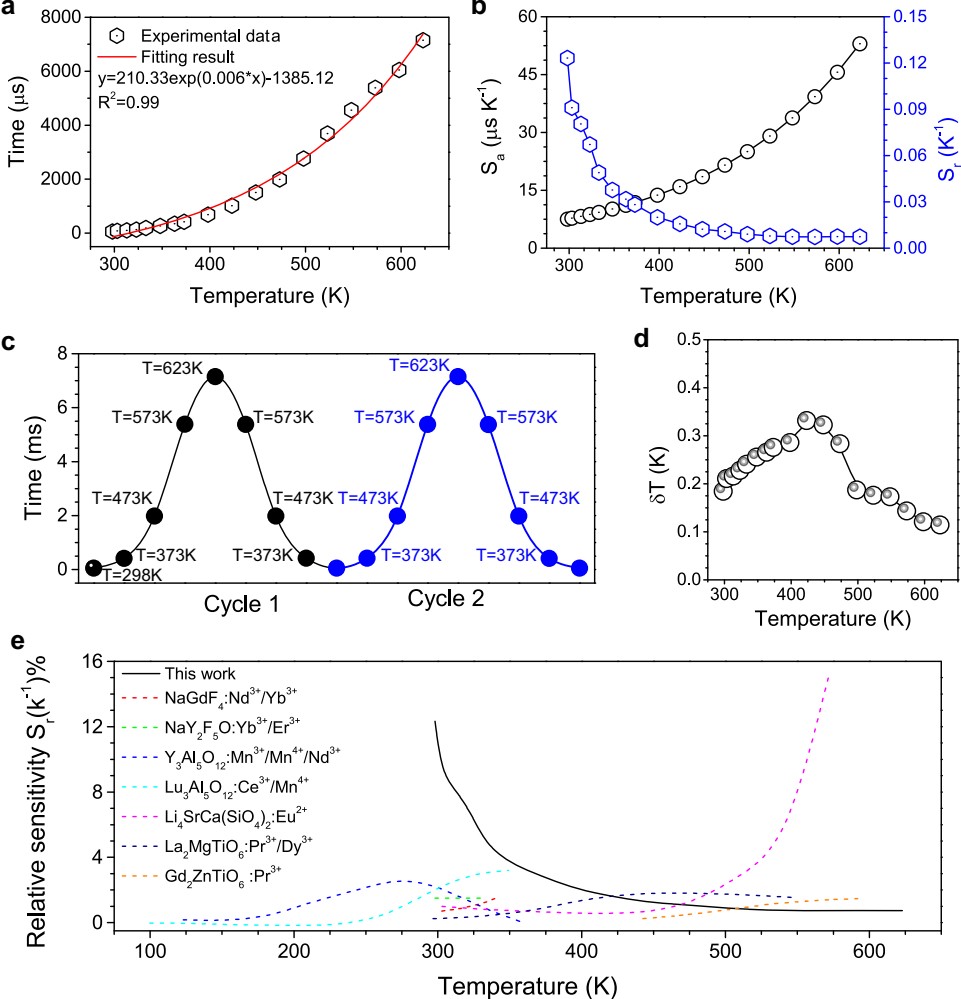

**Fig. 6 Lifetime-based luminescence thermometry. a** Experimentally measured and exponential fitted plots of lifetime $\tau$ of $Sc_2(MoO_4)_3$:20%Yb/1%Er at different temperatures. **b** Calculated $S_a$ and $S_r$ versus temperature based on the $Sc_2(MoO_4)_3$:20%Yb/1%Er. **c** Lifetime temperature-recycle measurements 2 cycles of heating and cooling between 298 and 623 K based on the $Sc_2(MoO_4)_3$:20%Yb/1%Er. **d** Temperature uncertainty for $Sc_2(MoO_4)_3$:20%Yb/1%Er at different temperatures. **e** Relative sensitivity ($S_r$) of $Sc_2(MoO_4)_3$:Yb/Er and the state-the-art luminescent thermometers. Different color dotted lines are previous work based on lifetime mode. $S_r$ are directly extracted from literature reports: $NaGdF_4$:$Nd^{3+}$/$Yb^{3+}$[47]; $Li_3SrCa(SiO_4)_2$:$Eu^{2+}$[48]; $NaY_2F_5O$:$Yb^{3+}$/$Er^{3+}$[49]; $Y_3Al_5O_{12}$:$Mn^{3+}$/$Mn^{4+}$/$Nd^{3+}$[50]; $Lu_3Al_5O_{12}$:$Ce^{3+}$/$Mn^{4+}$[51]; $La_2MgTiO_6$: $Pr^{3+}$/$Dy^{3+}$[52]; $Gd_2MgTiO_6$: $Pr^{3+}$[53].

For $Sc_2(MoO_4)_3$:20%Yb/1%Er phosphors, the plot of calculated lifetime value ($\tau$, μs) and temperature (T, K) is presented in Fig. 6a, where the dependence of lifetime on temperature can be well fitted by the following equation:

$$\tau(T) = -1385.12 + 210.33\exp 0.006T \tag{2}$$

Furthermore, the absolute temperature sensitivity ($S_a$) and relative temperature sensitivity ($S_r$) based on the luminescence lifetime of $Er^{3+}$ are calculated according to the following equations (Fig. 6b), respectively:[47]

$$S_a = \frac{d\tau}{dT} \tag{3}$$

$$S_r = \frac{1}{\tau} \cdot \frac{d\tau}{dT} \tag{4}$$

We have measured the temperature-dependent decay curves of $Sc_2(MoO_4)_3$:Yb/1%Er phosphors with different $Yb^{3+}$ concentrations (10%, 20%, and 25%). Accordingly, $Sc_2(MoO_4)_3$:20%Yb/1%Er phosphors exhibited the optimal $S_a$ of 53.0 μsK$^{-1}$ and $S_r$ of 12.3% K$^{-1}$ (Supplementary Fig. 14 and Table 1). Temperature-

recycle measurements demonstrated that the $Sc_2(MoO_4)_3$:20%Yb/1%Er can provide excellent thermal sensing repeatability (Fig. 6c). For comparison, we also displayed other state-of-the-art luminescent thermometry materials (Fig. 6e). The optimal $S_r$ value of $Sc_2(MoO_4)_3$:20%Yb/1%Er was observed to be an order of magnitude higher than that of other majority LLTs[47–53]. For example, the optimal $S_r$ value of $Sc_2(MoO_4)_3$:20%Yb/1%Er is higher than that (8.83% K$^{-1}$) of $SrTiO_3$:$Tb^{3+}$ nanocrystals[54].

The temperature uncertainty ($\delta$T) is an important parameter to assess the performance of a thermometer, which includes not only the relative sensitivity but also the error on the luminescence lifetime ($\delta\tau$)[55,56]. $\delta$T is calculated as follows:

$$\delta T = \frac{1}{S_r} \frac{\delta\tau}{\tau} \tag{5}$$

Where $\delta\tau/\tau$ is the uncertainty in the calculation of $\tau$ (determined as a standard deviation in 10 measurements of $\tau$, Supplementary Fig. 15), $S_r$ is the relative sensitivity of luminescence thermometer. Temperature uncertainty for $Sc_2(MoO_4)_3$:20%Yb/1%Er at different temperatures is presented in Fig. 6d. The minimum value of $\delta$T is 0.11 K even at 623 K. As such, despite the relatively low value of $S_r$ at the high-temperature range, the much longer

luminescence lifetime can effectively reduce the value of $\delta\tau/\tau$, resulting in lower $\delta T$. It should be noted that it is possible to keep the temperature uncertainty below a threshold of 0.33 K throughout the whole studied temperature range (298-623 K). The $\delta T$ threshold of $Sc_2(MoO_4)_3$:20%Yb/1%Er for LLT is much lower than that (0.7 K) of cubic-phase $LiLuF_4$:18%$Yb^{3+}$/2% $Er^{3+}$ nanocrystals for the conventional intensity-based thermometry[55]. All these results demonstrate that $Sc_2(MoO_4)_3$:Yb/Er phosphor can be explored as a kind of ideal lifetime-based luminescence thermometry with high $S_r$ and low $\delta T$.

## Discussion

In summary, a two-dimensional negative-thermal expansion phosphor of $Sc_2(MoO_4)_3$:Yb/Er has been proposed. The negative-thermal effect was systematically investigated by In situ temperature-dependent Raman spectroscopy, synchrotron X-ray diffraction and luminescence dynamics. We achieved 45-fold and 450-fold thermally enhanced UC luminescence and NIR-II DS emission of $Er^{3+}$ from 298 to 773 K, which is mainly benefitted by the thermally promoted energy transfer with increasing temperature. By virtue of the tunable PL lifetime of $Er^{3+}$ at different temperatures, we employed the proposed phosphor for LLT-based temperature sensing, which can circumvent the intrinsic limitations of poor temperature sensitivity based on the conventional intensity-based thermometry. These findings open up a new avenue for the exploration of thermal-enhanced luminescence phosphors with excellent UC and DS luminescence for versatile applications.

## Methods

**Chemical reagents**. All the chemicals of $Sc_2O_3$ (99.99%, Ganzhou Guangli High-tech Materials Co. Ltd), $Y_2O_3$ (99.99%, Ganzhou Guangli High-tech Materials Co. Ltd), $Yb_2O_3$ (99.99%, Ganzhou Guangli High-tech Materials Co. Ltd), $Er_2O_3$ (99.99%, Ganzhou Guangli High-tech Materials Co. Ltd), $(NH_4)_6Mo_7O_{24}\cdot4H_2O$ (AR, Shanghai Aladdin Biochemical Technology Co., Ltd.),HF ($\geq$ 40%, Xilong Scientific Co. Ltd), $HNO_3$, $C_6H_8O_7\cdot H_2O$ were used as raw materials.

**Synthesis of $Sc_2(MoO_4)_3$:Yb/Er**. A series of $Sc^{3+}$ substituted by different $Yb^{3+}$ ($Er^{3+}$) concentrations of $Sc_2(MoO_4)_3$:Yb/Er phosphors were synthesized by a sol-gel method[56]. Firstly, all the rare-earth nitrates were obtained by its oxides dissolving in dilute nitric acid. $(NH_4)_6Mo_7O_{24}\cdot4H_2O$ and $C_6H_8O_7\cdot H_2O$ were dissolved in a suitable volume of de-ionized water, respectively. The solution of $(NH_4)_6Mo_7O_{24}\cdot4H_2O$ and $C_6H_8O_7\cdot H_2O$ was slowly dropped into the solution containing rare-earth nitrates, into which a certain of citric acid as a chelating agent (citric acid/metal ion = 2:1) was added. The above solution was heated at 353 K to produce a light yellow transparent gel, which was further dried at 393 K for 24 h to get the yellow dried gel. Finally, the dried gel was annealed at 1073 K for 3 h in air. The white powder samples obtained via natural cooling were used for further characterization. Based on this synthetic method, a series of $Sc_{2(1-x-y)}(MoO_4)_3$:xYb/yEr (x = 0, 1%, 5%, 10%, 15%, 20%, 25%; y = 0, 0.1%, 0.5%, 1%, 2%, 4%, 8%) powders were prepared.

**Synthesis of $YF_3$:20%$Yb^{3+}$/1%$Er^{3+}$ phosphor**. $YF_3$:20%$Yb^{3+}$/1%$Er^{3+}$ phosphor was synthesized by co-precipitation method followed by heat treating in air atmosphere[57]. Firstly, 0.4460 g of $Y_2O_3$, 0.1970 g of $Yb_2O_3$ and 0.0096 g of $Er_2O_3$ were added to a quantity of $HNO_3$ and de-ionized water to form a rare earth nitrate solution. After all the rare earth oxides were dissolved, the HF solution was slowly added dropwise until the RE ions were completely precipitated and stirred for 30 min. The precipitate was filtered and washed three times with de-ionized water and anhydrous ethanol and dried in an oven at 353 K for 5 h. Finally, the white precipitate was calcined in a muffle furnace at 773 K for 1 h to obtain $YF_3$:20% $Yb^{3+}$/1%$Er^{3+}$ phosphor.

**Synthesis of $Er^{3+}$-doped $Yb_2WO_6$**. According to the stoichiometric composition of $Yb_{1.98}Er_{0.02}WO_6$, all the reactants were weighed and mixed thoroughly in an agate mortar, then sintered in a tubular furnace at 1573 K for 4 h in air. After cooled down to room temperature, the synthesized products were ground for subsequent analysis.

**Characterization**. The samples were characterized by powder X-ray diffraction (XRD) performed on a Panalytical X'Pert diffractometer using Cu K$\alpha$ radiation ($\lambda$ = 0.154187 nm). All of the patterns within the 10–90° 2$\theta$ range were collected in a scanning mode with a step size of 0.02°. The morphology patterns of samples were obtained on a field emission scanning electron microscope (SEM JSM-6700F) equipped with an Energy Dispersive Spectrometer spectra (EDS) and transmission electron microscope (TEM JEOL-2010). Thermogravimetric (TG) testing was made by using a NETZSCH STA2500 Regulus TG analyser. The $Sc_2(MoO_4)_3$:20%Yb/1%Er phosphor was placed in an alumina TG crucible on a TG sample tray and heated from 298 to 773 K at a heating rate of 5 K/min and mass loss was monitored during the heating process. The samples were purged with $N_2$ at 50 mL/min and 100 mL/min blow/sweep gas during the test. Nociolet 6700 Fourier transform infrared spectro-photometer (FTIR) with an MCT detector with low temperature was used for in situ temperature-dependent FTIR spectroscopy measurements. Spectra were obtained over the 4000–650 cm$^{-1}$ range for both sample and background single beam measurements.

**In situ Temperature-dependent synchrotron X-ray diffraction**. $Sc_2(MoO_4)_3$:20%Yb/1%Er phosphors were loaded into a corundum capillary with a diameter of 1.0 mm, which was then installed on the thermal stage of 14B1 beamline with a wavelength of 0.6887 Å (18 KeV) from Shanghai Synchrotron Radiation Facility. The detailed information about BL14B1 beamline was reported in the previous work[58]. To investigate the thermal expansion behavior of $Sc_2(MoO_4)_3$:20%Yb/1%Er phosphors, temperature-dependent synchrotron X-ray diffraction spectra were continuously measured as a set of circles on a two-dimensional image plate detector in the transmission mode during heating from 298 K to 673 K with the heating rate of 5 K min$^{-1}$. The total recording time for one spectrum was ~30 s.

**In situ temperature-dependent Raman spectroscopy**. In situ Raman spectroscopy was carried out using Renishaw inVia Reflex Raman microscope. Sample powders pressed into sheet were fixed in the heating device with $N_2$ as protective atmosphere. The laser with wavelength 532 nm was selected as the illuminant with ~1 μm diameter. The power of the laser was set below 1 mW/μm$^2$ to avoid sample damage. The acquisition time of each Raman spectrum was 12 s to 5 times for every 25 K from room temperature to 773 K. The laser spot was constantly adjusted through the microscope to make the beam stay at the same place during heating. The collected Raman spectra were fitted using Gauss and Voigt functions.

**In situ temperature-dependent photoluminescence**. The temperature-dependent UC (DS) luminescence spectra were evaluated by a FLS980 spectrophotometer equipped with a CW 980 nm diode laser (LASEVER INC. LSR980CP-FC-9W) as excitation source, and R928P (NIR-PMT) photo-multiplier as detector as well as a temperature control instrument (Linkam THMS600). The temperature-dependent luminescence decay curves were measured by a FLS980 instrument equipped with 980 nm pulse laser diode (laser: LASEVER INC. LSR980CP-FC-9W; signal generator: JDS6600). UC photographs at various temperatures were taken by the visible Nikon D750 camera under CW 980-nm laser (power density: 0.57 W/cm$^2$). To avoid interference of the 980-nm laser, the GRB3 filter was added in front of the visible camera. NIR photographs of $Sc_2(MoO_4)_3$:20%Yb/1%Er at various temperatures were taken by a NIR camera (Photonic Science). To avoid interference of the 980-nm laser (power density: 0.32 W/cm$^2$), the 1250 nm long-pass filter was added in front of the NIR camera.

## Data availability

The data that support the findings of this study are available in the Source Data. Source data are provided with this paper.

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

## Acknowledgements

JXUST's work was financially supported by the National Natural Science Foundation of China (No. 51862012), Jiangxi Provincial Natural Science Foundation (No. 20202BAB204008), National key research and development program (2020YFB1713700) and Jiangxi Provincial Key Laboratory of Functional Molecular Materials Chemistry (20212BCD42018) and the Innovation Leadership Program of Ganzhou. FIRSM's work was financially supported by the National Natural Science Foundation of China (No. U1805252) and the CAS/SAFEA International Partnership Program for Creative Research Teams. The authors thank beamline BL14B1 (Shanghai Synchrotron Radiation Facility) for providing the beam time and assistance.

## Author contributions

J.L. conceived the idea and designed the research. M.W., Z.H., and B.F. synthesized and characterized materials. F.L. performed temperature-dependent luminescence spectra. J.L., D.T., X.C., and B.Q. analyzed the data and wrote the paper. H. W. supervised all the research. All the authors contributed to the discussion and provided feedback on the manuscript.

## Competing interests

The authors declare no competing interests.
