## [Peer Review File · Nature Communications]

nature portfolio

Peer Review File

Draft OnlyREVIEWER COMMENTS

Reviewer #1 (Remarks to the Author):

The work concerns a very interesting and popular topic - luminescence thermometry. The presented strategy using a material with a negative temperature expansion coefficient as a material with a positivistic thermal luminescence coefficient seems to be a very promising approach offering very high relative sensitivities. Therefore, I believe that the work has an important aspect of novelty. I am not sure that it will be of interest to a wide readership of this journal. Nevertheless, some issues should be clarified before the work can be considered for publication in Nature Communications:

1. Caption of Figure 6 is misleading. Actually, the authors do not show any application there.
2. Authors should, in accordance with current standards, use an uppercase "K" to designate temperature units
3. Although the summary of results presented in Figure 6d is very informative, I encourage the authors to contrast the results presented with respect to recently published work for which very high relative sensitivities were obtained such as [10.1016/j.cej.2021.131165](https://doi.org/10.1016/j.cej.2021.131165)
4. The interionic distance contraction is one of the very specific mechanisms responsible for the thermal enhancement of the luminescence. There are several other processes that may lead to the similar thermal behaviors like i.e. change of the local ion symmetry etc.
5. The purpose of the data shown in Figures 1 a, b, c is to describe three types of temperature effect trends on luminescence intensity. Therefore, I would suggest removing the "diagrams" placed under the graphs to maintain the universality of the description presented.
6. Authors should use consistent temperature units throughout the paper (e.g. Fig.1 Celsius and Figure 6 Kelvins).
7. Please remove the information about the wavelength of the radiation used in the brackets of the axis captions in Figures 2a and b
8. The increase in emission intensity with increasing temperature may be due to an increase in the absorption cross section associated with the spectral broadening of the absorption band. This occurs when the excitation line etches into the slope of the absorption band. Therefore, the authors should present the excitation spectra of Yb³⁺ ions measured as a function of temperature

9. The magnitude of the thermal contraction of the crystallographic unit cell depends on the difference in the ionic radii between dopants and replaces ions and on the concentration of dopants. Therefore, I would like the authors to comment if it is possible to modulate the thermometric parameters of this type of thermometer by changing the dopant concentration?

10. Table S1- the comparing the thermometric properties of the luminescent thermometry operating in different readout modes is unjustified and unreasonable.

Reviewer #2 (Remarks to the Author):

The authors of “Thermally boosted upconversion and downshifting luminescence in $\text{Sc}_2(\text{MoO}_4)_3:\text{Yb}/\text{Er}$ with two-dimensional negative thermal expansion” presented a very interesting research of the famous upconverting pair codoped in more than astounding host matrix. However, for the following reasons I recommend major revision and reinspection, as there are numerous problems with novelty, significance and support for the conclusions. Briefly these problems can be summed as:

- The same host matrix was investigated for thermometry by the Yb/Ho pair and LIR with similar conclusions. This just slightly diminishes the stated novelty.
- The discussion about inter-atomic differences is possibly erroneous, and need either serious revision or removal.
- The overall intensity of this phosphor is unknown. For this the authors must expand their research as suggested.
- This high relative sensitivity occurs at the points where the signal is borderline detectable. This means that the temperature resolutions will be low at those points, i.e. the significance of the research is not as high as initially suggested. The authors need to include temperature resolution discussion in the most correct manner, and accordingly adjust their conclusions.

Issues in detail:

In the first reading the novelty of this article seems immense. However, the exactly the same host, co-doped with an upconverting pair of $\text{Yb}^{3+}/\text{Ho}^{3+}$ has been already explored for luminescence thermometry in Ref: <https://pubs.acs.org/doi/10.1021/acs.jpcllett.0c00628> . Although the authors did reference this work, their contribution and similarity needs to be more stressed. That being said, the differences in employing the more important upconverting pair, $\text{Yb}^{3+}/\text{Er}^{3+}$, and lifetime instead

of luminescence intensity ratio method, should be stated as the novelty of this work. In my opinion, these subtle differences are important and enough novelty for the publication of this work.

For the equation 1 the referenced is the famous short work by Blasse (Ref. 22). However, in that reference the Blasse is not reporting on a way of calculation of the average distance, but on the critical distance, i.e. when the concentration quenching starts to dominate. The c in the subscript of x_c does not abbreviate concentration, but the word critical. Up to my knowledge, this equation cannot be applied for calculation of distance between interacting ions if that distance is not critical distance, and even more, of the ions of different types. Unless the authors can provide another, a very strong reference to support their claims of usage of equation 1, I suggest removing the Figure 2d, Equation 1, and any corresponding discussion.

There is another issue with concentrations and distances. In figure 2 no concentrations of dopants are mentioned and yet there is a graph of their distances. To additionally confirm the assumption that the author's analysis of interatomic distances is incorrect is the un-logical trends in presented distances. Although both the codopants substitute the same ions in the SMO matrix, the authors claim that Yb-Yb and Yb-Er distances decrease with temperature, but the Er-Er distances increase. All three claims cannot be possible, if I understood well, if not, then the paragraph needs to be revised for clarity. As for the red emission of Er³⁺, the authors should see the book by Kaminskii, Crystalline lasers, where the pathways for population of the 4F_{9/2} are explained, and they are related to the Yb³⁺ ion back-transfer more than the Er³⁺ cross-relaxation.

Lifetime method for thermometry is erroneously described as ratiometric multiple times in the abstract and throughout the article. The definition of ratiometric fluorescence is where intensities of at least two bands are measured and compared. Thus, the word ratio in ratiometric. FIR or LIR methods, of which the authors are certainly aware of, are the ratiometric luminescence thermometry methods, while lifetime, although also self-referenced, is not. This needs to be corrected in the caption of Table S1 as well.

Throughout the article there needs to be more linear story of using concentrations and even hosts. It is hard to follow which concentrations are being used, and when it is compared to the other host. For each spectra the precise concentrations of codopants should be given.

The authors discuss the 4I_{11/2} emission of Er³⁺, however in the spectra and in the text the 1538 nm emission is mentioned. Did you mean the emission from 4I_{13/2} instead, as the emission from 4I_{11/2} to the ground level is at about 1000 nm? See for example Figure S9b: caption says 4I_{11/2} and $\lambda_{em} = 1538$ nm, the same in Figure 5.

For other journals of lower esteem than the Nature series, the comparison with other hosts would not be necessary. However, I feel that in this host the emission intensity is much lower than in other famous upconverting hosts, for example YF₃ or NaYF₄. I would like to see the comparison in either intensities or quantum yields between SMO and one of these prominent hosts.

Lastly but not the least, this amazing sensitivity occurs at the exact same temperatures where the emission intensities are very low. Thus, at the points of the highest relative sensitivity, the sensing might be just possible, as the uncertainty in measurement will be great. This was discussed in the article: <https://onlinelibrary.wiley.com/doi/full/10.1002/adts.202000176>. This last point

significantly diminishes the firstly claimed importance of the obtained results. I invite the authors to be honest in reporting the sensitivities only on the practically usable range. Also, taking everything in consideration, there is a must that the authors report on the temperature resolutions at the whole temperature range, and unfortunately this means reporting on the uncertainties that should be evaluated by at least several temperatures. Uncertainties should be given as the standard deviation of at least 10 consecutive measurements at the same temperature. This temperature resolutions will show the true usability of the sensor, and my assumption is that the temperature resolution curve will be different than the sensitivity curves.

Reviewer #3 (Remarks to the Author):

The authors present a two-dimensional negative thermal phosphor SMO:Yb/Er systematic study. They present a temperature dependent Raman, synchrotron X-ray diffraction and luminescence investigation. The thermally boosted UC and DS luminescence mechanism was investigated. The authors claim that the luminescence lifetime temperature dependency enables ratiometric thermometry with high relative sensitivity; an important topic in current research.

This is a comprehensive work, very well written and with insights into the Ln³⁺-doped phosphors for potential applications.

Although they address an important topic some issues, that I refer below, need further clarification.

Authors should comment, if known, what is the mechanism for the NTE behaviour in this system, whether it is structural, or electronically based for instance?

What is supposed to drive the transition from PTE to NTE around 100°C?

The transition from PTE to NTE occurs at ca 100°C (figure 2c). How this transition impacts the luminescence properties and how this is compatible with the suggested thermally enhanced photoluminescence mechanism. It is important to clarify this issue.

Some Raman modes appear to alter above 100 K, can they be related with the NTE to PTE transition?

From lines 182-183: "Moreover, the distance of Yb³⁺-Er³⁺ becomes shorter with an increase of the temperature. The ET processes between sensitizer (Yb³⁺) to activator (Er³⁺) are usually considered to occur through dipolar-dipolar interactions, whose ET efficiency is proportional to r^{-6} (r is the average donor-acceptor distance). The result indicates that the distances of Yb³⁺-Er³⁺ ions (13.975 Å, at 25 °C) decreased at higher temperature, which may benefit the improvement of the ET efficiency between Yb³⁺-Er³⁺."

And line 236 "Based on the above analysis, the thermally enhanced UC emission with the temperature from 25 to 200 °C is mainly governed by the ET from Yb³⁺ to Er³⁺"

Authors should clarify the nature for thermally enhanced UC emission within the temperature range from 25 to 200 °C, if this effect is mainly governed by the ET from Yb³⁺ to Er³⁺, how it relates with the PTE of the system shown in Fig. 2 observed between 25 °C and 100 °C.

From lines 46-49: " Most of such abnormal thermally enhanced UC luminescence is observed in Ln³⁺-doped inorganic materials with three-dimensional negative-thermal expansion characteristics (NTE)^{9, 10}, where all the three cell parameters of the doped crystals shrink at elevated temperature."

And below, lines 58-59 "with a unique two-dimensional NET coefficient ($\alpha_a = -8.62 \times 10^{-6}/K$, $\alpha_b = 4.25 \times 10^{-6}/K$, $\alpha_c = -6.35 \times 10^{-6}/K$) "

In the context of this work, is there any effective advantage in two-dimensional negative-thermal expansion materials when compared with the three-dimensional ones? The authors could also mention recent works with uniaxial NTE.

Can the authors explain in what way the Energy migration between Ln³⁺ dopants is confined to two dimensions?

From lines 231-233. The promoted Ar and inhibited Anr in the Ln³⁺ dopant indicate that lattice distortion enhances a crystal field with odd parity and modifies local symmetry of activator ions via the NTE effect.

The authors mention the existence of lattice distortion and a change in the local symmetry of activator ions via the NTE effect. Can these local distortions be substantiated via the temperature-dependent Raman data?

Can this effect be witnessed by the other techniques that would perceive the negative thermal expansion at the local scale? The authors could comment on that.

Along the manuscript:

What is the degree of Sc substitution in the samples that were measured in the presented results Figure 2,3,4,5? This is only specified in the labels of the Figures presented in the Supplementary material.

line 18 "decrease at the higher temperature" author should substitute "decrease for high temperatures"

line 58 and 103 Where is the abbreviation "NET" defined? Do the authors mean "NTE"?

Line 121-122: What is the value of X_c used for the calculations?

Line 133 -135 "The observed mode with a frequency of 341 cm^{-1} corresponds to the bending mode of MoO_4 tetrahedra. When the temperature increases, this Raman peak exhibits a blue shift (Fig. 2f). The median frequency B1g mode of SMO is observed at 511 cm^{-1} "

Is this a typo? Figure 2f is not from the 341 cm^{-1} mode. The sentence should be improved.

The authors refer the studied systems as "the structures of the SMO:Yb/Er samples with different dopant concentrations", doping generally refers to a small concentrations, here the degree of ionic Sc substitution by Yb and Er cations reaches the high value of 25% or 8%. The term chemical substitution would be more appropriated.

Reply to the Comments of Reviewer #1

General comment:

The work concerns a very interesting and popular topic - luminescence thermometry. The presented strategy using a material with a negative temperature expansion coefficient as a material with a positivistic thermal luminescence coefficient seems to be a very promising approach offering very high relative sensitivities. Therefore, I believe that the work has an important aspect of novelty. I am not sure that it will be of interest to a wide readership of this journal. Nevertheless, some issues should be clarified before the work can be considered for publication in Nature Communications:

Response:

We greatly appreciate the reviewer for his/her positive comments on our manuscript and the efforts to improve the quality of our manuscript. A point-by-point response is noted below.

Comment #1:

Caption of Figure 6 is misleading. Actually, the authors do not show any application there.

Response:

Thank you for your kind reminding. Caption of Figure 6 has been corrected as “Lifetime-based luminescence thermometry” in the revised manuscript. Please see also line 1 of p. 15 in the revised manuscript.

Comment #2:

Authors should, in accordance with current standards, use an uppercase "K" to designate temperature units

Response:

Temperature units have been revised as uppercase "K" in the revised manuscript.

Comment #3:

Although the summary of results presented in Figure 6d is very informative, I encourage the authors to contrast the results presented with respect to recently published work for which very high relative sensitivities were obtained such as 10.1016/j.cej.2021.131165.

Response:

The suggested reference has been cited as Ref. 54, which was compared with our results and discussed in the revised manuscript. Specifically, the optimal S_r value of SMO:20%Yb/1%Er ($12.3\%K^{-1}$) is higher than that ($8.83\% K^{-1}$) of SrTiO₃:Tb³⁺ nanocrystals in the suggested reference (*Chem. Eng. J.* **2022**, 428, 131165). Please see also line 27 of p.14 in the revised manuscript.

Comment #4:

The interionic distance contraction is one of the very specific mechanisms responsible for the thermal enhancement of the luminescence. There are several other processes that may lead to the similar thermal behaviors like i.e. change of the local ion symmetry etc.

Response:

Many thanks for this valuable suggestion. Indeed, the interionic distance contraction of the different axis is one of the very specific mechanisms responsible for the thermal enhancement of luminescence. Besides, the change of the local ion symmetry may also affect the optical properties of RE³⁺ ions like luminescence intensity and radiative transition. In this work, we propose a new class of NTE phosphors based on SMO:Yb/Er exhibiting a unique two-dimensional NTE coefficient. With increasing the temperature, the SMO:Yb/Er phosphors experience an anisotropic two-dimensional shrinkage along a and c axes at elevated temperature, whereas the distance between RE³⁺-RE³⁺ ions increases along b axis. As such, the detrimental energy migration of RE³⁺-RE³⁺ ions can be effectively confined within the two-dimensional layered structures (e.g., (010) lattice plane) at elevated temperature. More discussion has been added. Please see also lines 7-16 of p. 5 in the revised manuscript.

Comment #5:

The purpose of the data shown in Figures 1 a, b, c is to describe three types of temperature effect trends on luminescence intensity. Therefore, I would suggest removing the "diagrams" placed under the graphs to maintain the universality of the description presented.

Response:

Thanks a lot. We have removed the "diagrams" placed under the graphs in the revised manuscript. The captions of Figures 1 a, b, c have been corrected as “ **Figure 1| Scheme of thermal-dependence effects in phosphor. a. Positive thermal quenching phenomenon. b. Zero thermal quenching phenomenon. c. Negative-thermal quenching (thermal-enhanced) phenomenon.**”. Please see also line 1 of p. 3 in the revised manuscript.

Comment #6:

Authors should use consistent temperature units throughout the paper (e.g. Figure 1 Celsius and Figure 6 Kelvins).

Response:

All the temperature units have been corrected as **Kelvins** in the revised manuscript.

Comment #7:

Please remove the information about the wavelength of the radiation used in the brackets of the axis captions in Figures 2a and b.

Response:

Thanks. The information about the wavelength of the radiation used in the brackets of the axis captions in Figures 2a and 2b has been removed in the revised manuscript. Please see also line 1 of p. 6 in the revised manuscript.

Comment #8:

The increase in emission intensity with increasing temperature may be due to an increase in the absorption cross section associated with the spectral broadening of the absorption band. This occurs when the excitation line etches into the slope of the absorption band. Therefore, the authors should present the excitation spectra of Yb^{3+} ions measured as a function of temperature.

Response:

Many thanks for this valuable suggestion. We have added the excitation spectra of Yb^{3+} ions measured as a function of temperature in the revised manuscript (Supplementary Figure 5 newly added). Accordingly, the absorption band centered at 980 nm was hardly broadened with increasing temperature. The intensity of the excitation peak increased with increasing the temperature from 298 to 573 K, which may benefit the luminescence emission of Yb^{3+} at the higher temperature. Please see also lines 7-11 of p. 8 in the revised manuscript.

Supplementary Figure 5| Temperature-dependent excitation spectra of SMO:20%Yb/1%Er phosphor.

Comment #9:

The magnitude of the thermal contraction of the crystallographic unit cell depends on the difference in the ionic radii between dopants and replaces ions and on the concentration of dopants. Therefore, I would like the authors to comment if it is possible to modulate the thermometric parameters of this type of thermometer by changing the dopant concentration?

Response:

Many thanks for this valuable suggestion. It is true that the magnitude of the thermal contraction of the crystallographic unit cell depends on the difference in the ionic radii between dopants and replaces ions and on the concentration of dopants. Therefore, we investigated the thermometric parameters of SMO:xYb/1%Er with different concentrations of sensitizer (Yb^{3+}). Following the reviewer's suggestion, we have measured the temperature-dependent decay curves of SMO:Yb/1%Er and phosphors with different Yb^{3+} concentrations (10%, 20% and 25%), which were newly added as Supplementary Figure 14 in the Supporting Information. **As indicated in Supplementary Table 1, SMO:20%Yb/1%Er phosphors exhibit the optimal absolute sensitivity S_a ($53.0 \mu\text{sK}^{-1}$) and relative sensitivity S_r ($12.3\% \text{K}^{-1}$).** Please see also lines 20-23 of p. 14 in the revised manuscript.

Supplementary Figure 14 | Lifetime-based luminescence thermometry in $x\%Yb^{3+}/1\%Er^{3+}$ -codoped SMO with different Yb^{3+} concentrations. a. Temperature-dependent luminescence decay curves of $^4I_{13/2}$ excited states of Er^{3+} in $10\%Yb^{3+}/1\%Er^{3+}$ -codoped SMO. **b.** Experimentally measured and exponentially fitted plots of lifetime τ of SMO:10%Yb/1%Er at different temperatures. **c.** Calculated absolute sensitivity S_a and relative sensitivity S_r versus temperature based on the SMO:10%Yb/1%Er. **d.** Temperature-dependent luminescence decay curves of $^4I_{13/2}$ excited states of Er^{3+} in $25\%Yb^{3+}/1\%Er^{3+}$ -codoped SMO. **e.** Experimentally measured and exponentially fitted plots of lifetime τ of SMO:25%Yb/1%Er at different temperatures. **f.** Calculated absolute sensitivity S_a and relative sensitivity S_r versus temperature based on the SMO:25%Yb/1%Er.

Supplementary Table 1|. Lifetime-based luminescence thermometry parameters in x% Yb³⁺/1% Er³⁺-codoped SMO with different Yb³⁺ concentrations.

Sample	S _a (μs/K)	S _r (%/K)
10% Yb/1% Er:SMO	31.8	3.0
20% Yb/1% Er:SMO	53.0	12.3
25% Yb/1% Er:SMO	31.2	7.1

Comment #10:

Table S1- the comparing the thermometric properties of the luminescent thermometry operating in different readout modes is unjustified and unreasonable.

Response:

Thank you. Table S1 has been deleted in the revised manuscript.

Draft Only

Reply to the Comments of Reviewer #2

General comment:

The authors of “Thermally boosted upconversion and downshifting luminescence in $\text{Sc}_2(\text{MoO}_4)_3:\text{Yb/Er}$ with two-dimensional negative thermal expansion” presented a very interesting research of the famous upconverting pair codoped in more than astounding host matrix. However, for the following reasons I recommend major revision and reinspection, as there are numerous problems with novelty, significance and support for the conclusions.

Response:

We greatly appreciate the reviewer for his/her positive comments on our manuscript and the efforts to improve the quality of our manuscript. We have made all the requested changes in the revised manuscript according to the reviewer's comments. Particularly, more control experiments and analyses have been added to highlight the novelty of our work. A point-by-point response is noted below.

Comment #1:

The same host matrix was investigated for thermometry by the Yb/Ho pair and LIR with similar conclusions. This just slightly diminishes the stated novelty.

Response:

Thank you for pointing out this important issue. It is true that the same host matrix was investigated as traditional LIR thermometry by the Yb/Ho pair (Ref. 10). **Nevertheless, the content and novelty of our work are totally different from the previous report.** For the first time, we have revealed the unique two-dimensional NTE of $\text{SMO}:\text{Yb/Er}$ phosphors, which exhibited simultaneous enhancements of upconversion and downshifting photoluminescence (PL) of Er^{3+} by 45-fold and 450-fold from 298 to 773 K, respectively. Besides, the strategy of NIR downshifting luminescence lifetime instead of the conventional visible UC luminescence intensity ratio method was applied for temperature sensing. The near-infrared luminescence lifetime of Er^{3+} in $\text{SMO}:\text{Yb/Er}$ was determined to span two orders of magnitude from 298 to 623 K, which enables their application in lifetime-based luminescent thermometry with high absolute temperature sensitivity and relative temperature sensitivity. **Our work presents a substantial advance as compared to previous work and thus could be a breakthrough in the development of RE^{3+} -doped NTE phosphors.** Please see also lines 20-26 of p. 16 in the revised manuscript.

Comment #2:

The discussion about inter-atomic differences is possibly erroneous, and need either serious revision or removal.

Response:

Many thanks for this valuable suggestion. We have corrected the discussion about inter-atomic differences, which were estimated by the crystallographic information file (CIF) of Rietveld refinement of the *in situ* temperature-dependent SXRD. Please see also lines 7-16 of p. 5 in the revised manuscript.

Comment #3:

The overall intensity of this phosphor is unknown. For this the authors must expand their research as suggested.

Response:

Many thanks for this valuable suggestion. We have added the discussion about the overall intensity of this phosphor. To explicitly indicate the overall intensity of this phosphor, the famous Orthorhombic-phase $\text{YF}_3:20\% \text{Yb}/1\% \text{Er}$ phosphor with the same crystallographic system of SMO is selected as the control samples. To compare the emission intensity of $\text{YF}_3:20\% \text{Yb}/1\% \text{Er}$ and $\text{SMO}:20\% \text{Yb}/1\% \text{Er}$ phosphor, we have measured the temperature-dependent UC/DS luminescence spectra of $\text{YF}_3:20\% \text{Yb}/1\% \text{Er}$ and $\text{SMO}:20\% \text{Yb}/1\% \text{Er}$ under otherwise identical conditions (Supplementary Figures. 6 and 7 newly added). It can be observed that the UC/DS intensity of $\text{YF}_3:20\% \text{Yb}/1\% \text{Er}$ phosphor decreased continuously with increasing temperature, while the UC/DS intensity of $\text{SMO}:20\% \text{Yb}/1\% \text{Er}$ phosphor increased markedly with increasing the temperature. Specifically, the overall UC/DS intensity of $\text{YF}_3:20\% \text{Yb}/1\% \text{Er}$ is much higher than that of $\text{SMO}:20\% \text{Yb}/1\% \text{Er}$ at 298 K. Nevertheless, the overall UC and DS intensity of $\text{SMO}:20\% \text{Yb}/1\% \text{Er}$ phosphor are 4.5 and 12.9 times higher than that of $\text{YF}_3:20\% \text{Yb}/1\% \text{Er}$ counterpart at 773 K, respectively. **These results explicitly validate the superiority of the SMO:Yb/Er as novel luminescent materials over existing PTE phosphors, particularly at high temperatures.** The above discussions have been added in lines 1-13 of p. 10 and Supplementary Figures 6-7 newly added in the Supporting Information.

Comment #4:

This high relative sensitivity occurs at the points where the signal is borderline detectable. This means that the temperature resolutions will be low at those points, i.e. the significance of the research is not as high as initially suggested. The authors need to include temperature resolution discussion in the most correct manner, and accordingly adjust their conclusions.

Response:

Thank you for pointing out this important issue. To determine the relative sensitivity of our phosphors at other temperatures, we have measured the luminescence lifetimes of our phosphors at four temperature points (303, 313, 333, 363 K). All the PL decays

for each temperature point were measured independently for ten times under identical conditions to yield the average value. Temperature resolution discussion via data analysis of the 10 consecutive measurements has been newly added in lines 1-14 in p. 16 in the revised manuscript. A slight change in sensitivity was obtained by re-fitting the data by adding data pointing. From Figures 5 a and 5b, it is obvious that the data become dense at the temperature range from 298 to 373K, which may improve the data accuracy via decreasing the fitting error. The optimal S_a and S_r were determined to be as high as $53.0 \mu\text{K}^{-1}$ and $12.3\% \text{ K}^{-1}$, respectively. The fitting results have been adjusted in the revised manuscript.

Comment #5:

Issues in detail:

In the first reading the novelty of this article seems immense. However, the exactly the same host, co-doped with an upconverting pair of $\text{Yb}^{3+}/\text{Ho}^{3+}$ has been already explored for luminescence thermometry in Ref: <https://pubs.acs.org/doi/10.1021/acs.jpcllett.0c00628>. Although the authors did reference this work, their contribution and similarity needs to be more stressed. That being said, the differences in employing the more important upconverting pair, $\text{Yb}^{3+}/\text{Er}^{3+}$, and lifetime instead of luminescence intensity ratio method, should be stated as the novelty of this work. In my opinion, this subtle differences are important and enough novelty for the publication of this work.

Response:

Many thanks for this valuable suggestion. Previously, SMO:Yb/Ho with same host was used as a ratiometric thermometer based on the UC red-to-green emission intensity ratio. Their contribution and similarity about the same SMO:Yb/Ho used as luminescence thermometry in Ref. 10 have been stressed in the revised manuscript. The above discussions have been added in lines 29-30 of p. 13 in the revised manuscript.

It should be noted that the content and novelty in this work are totally different from the previous report. **For the first time, we have revealed the unique two-dimensional NTE of SMO:Yb/Er phosphors, which exhibited simultaneous enhancements of upconversion and downshifting PL of Er^{3+} by 45-fold and 450-fold from 298 to 773 K, respectively.** Besides, the strategy of NIR downshifting luminescence lifetime instead of the conventional visible UC luminescence intensity ratio method was applied for temperature sensing. The near-infrared luminescence lifetime of Er^{3+} in SMO:Yb/Er was determined to span two orders of magnitude from 298 to 623 K, which enables their application in lifetime-based luminescent thermometry with high absolute temperature sensitivity and relative temperature sensitivity. **Our work presents a substantial advance as compared to previous work and thus could be a breakthrough in the development of RE^{3+} -doped NTE**

phosphors. We sincerely hope the reviewer concurs after reading this clarification.

Comment #6:

For the equation 1 the referenced is the famous short work by Blasse (Ref. 22). However, in that reference the Blasse is not reporting on a ways of calculation of the average distance, but on the critical distance, i.e. when the concentration quenching starts to dominate. The *c* in the subscript of *xc* does not abbreviate concentration, but the word critical. Up to my knowledge, this equation cannot be applied for calculation of distance between interacting ions if that distance is not critical distance, and even more, of the ions of different types. Unless the authors can provide another, a very strong reference to support their claims of usage of equation 1, I suggest removing the Figure 2d, Equation 1, and any corresponding discussion.

Response:

Many thanks for this valuable suggestion. Following the reviewer's suggestion, we have updated Figure 2d. For the case of homogeneous substitution of Sc^{3+} by RE^{3+} ions in SMO, the changing trend of the distance of RE^{3+} - RE^{3+} ions ($\text{RE} = \text{Sc}/\text{Yb}/\text{Er}$) along different crystal axis-direction in the structure diagram for different temperatures will reflect the changing trend of the distance between the Yb^{3+} and Er^{3+} ions. With the increase of temperature, the distance of RE^{3+} - RE^{3+} ions along *a*-axis (6.7061 Å, at 298 K) contracts, and the distance of RE^{3+} - RE^{3+} ions along *b*-axis (10.8288 Å, at 298 K) expands steadily, while the distance of RE^{3+} - RE^{3+} ions along *c*-axis first expands in the temperature range (298-373 K) and then contracts. The results indicate that the distances of RE^{3+} - RE^{3+} ions along *a/c*-axis decreased at higher temperatures, which may benefit the improvement of the energy-transfer efficiency between Yb^{3+} and Er^{3+} . See also the updated discussion in lines 7-16 on page 5 of the revised manuscript.

Figure 2. Temperature-dependent proximate distances of RE-RE ($\text{RE} = \text{Sc}/\text{Yb}/\text{Er}$) along different axes derived from the unit cell structure.

Comment #7:

There is another issue with concentrations and distances. In figure 2 no concentrations of dopants are mentioned and yet there is a graph of their distances. To additionally confirm the assumption that the author's analysis of interatomic distances is incorrect is the un-logical trends in presented distances. Although both the codopants substitute the same ions in the SMO matrix, the authors claim that Yb-Yb and Yb-Er distances decrease with temperature, but the Er-Er distances increase. All three claims cannot be possible, if I understood well, if not, then the paragraph needs to be revised for clarity. As for the red emission of Er³⁺, the authors should see the book by Kaminskii, Crystalline lasers, where the pathways for population of the ⁴F_{9/2} are explained, and they are related to the Yb³⁺ ion back-transfer more than the Er³⁺ cross-relaxation.

Response: Thank you for pointing out our mistakes. In Figure 2, the Yb³⁺/Er³⁺ dopant concentrations are 20% and 1%, respectively, which was added in the revised manuscript.

For the incorrect description “Yb-Yb and Yb-Er distances decrease with temperature, but the Er-Er distances increase.”, we have updated Figure 2d, wherein the distances of RE³⁺-RE³⁺ ions at different planes were determined by the crystallographic information file (CIF) of *Rietveld* refinement on the basis of the SXRD patterns (Supplementary Figure 3c). See also the updated discussion in lines 1-10 on page 6 of the revised manuscript.

The confusion of the temperature-dependent luminescence mechanism in this work was perhaps caused by our unclear writing in the manuscript. Regarding the red emission of Er³⁺, it is true that the shortened Yb³⁺-Er³⁺ interatomic distance may contribute to the back energy transfer from Er³⁺ to Yb³⁺ ions: ⁴S_{3/2} (Er³⁺) + ²F_{7/2} (Yb³⁺) → ⁴I_{13/2} (Er³⁺) + ²F_{5/2} (Yb³⁺), followed by energy transfer from Yb³⁺ to Er³⁺ through ⁴I_{13/2} (Er³⁺) + ²F_{5/2} (Yb³⁺) → ⁴F_{9/2} (Er³⁺) + ²F_{7/2} (Yb³⁺). As a result, the population of the ⁴F_{9/2} states is enhanced to produce stronger red UC emissions (Ref. 34 Kaminskii A., *Crystalline Lasers: Physical Processes and Operating Schemes*. (CRC Press, 2020).). The above discussion has been added in lines 20-24 of p. 7 in the revised manuscript.

Comment #8:

Lifetime method for thermometry is erroneously described as ratiometric multiple times in the abstract and throughout the article. The definition of ratiometric fluorescence is where intensities of at least two bands are measured and compared. Thus, the word ratio in ratiometric. FIR or LIR methods, of which the authors are certainly aware of, are the ratiometric luminescence thermometry methods, while lifetime, although also self-referenced, is not. This needs to be corrected in the caption of Table S1 as well.

Response: Thanks. We have carefully checked the manuscript and revised similar mistakes. In the abstract “the luminescence lifetime of ${}^4I_{11/2}$ of Er^{3+} in SMO:Yb/Er displays a strong temperature dependence, enabling **radiometric** thermometry with the highest relative sensitivity of 12.3%/K at 298 K” has corrected as “the luminescence lifetime of ${}^4I_{11/2}$ of Er^{3+} in SMO:Yb/Er displays a strong temperature dependence, enabling **luminescence** thermometry with the highest relative sensitivity of 12.3%/K at 298 K”. Please see also line 30 p. 1 in the revised manuscript. Besides, Table S1 has been deleted according to the comment of the first reviewer.

Comment #9:

Throughout the article there needs to be more linear story of using concentrations and even hosts. It is hard to follow which concentrations are being used, and when it is compared to the other host. For each spectra the precise concentrations of codopants should be given.

Response:

Thank you for your kind reminding. For each spectrum, the precise concentration of dopants has been given in the revised manuscript.

Comment #10:

The authors discuss the ${}^4I_{11/2}$ emission of Er^{3+} , however in the spectra and in the text the 1538 nm emission is mentioned. Did you mean the emission from ${}^4I_{13/2}$ instead, as the emission from ${}^4I_{11/2}$ to the ground level is at about 1000 nm? See for example Figure S9b: caption says ${}^4I_{11/2}$ and $E_{em}=1538$ nm, the same in Figure 5.

Response:

Thank you for your kind reminding. Indeed, the emission peak of 1538 nm originated from ${}^4I_{13/2}$ of Er^{3+} . All the “ ${}^4I_{11/2}$ ” have been revised as “ ${}^4I_{13/2}$ ” in the revised manuscript.

Comment #11:

For other journals of lower esteem than the Nature series, the comparison with other hosts would not be necessary. However, I feel that in this host the emission intensity is much lower than in other famous upconverting hosts, for example YF3 or NaYF4. I would like to see the comparison in either intensities or quantum yields between SMO and one of these prominent hosts.

Response:

Many thanks for this valuable suggestion. Following the reviewer's suggestion, we have selected YF₃:20%Yb/1%Er phosphor as the control group. As we know, the

famous Orthorhombic-phase $\text{YF}_3:20\% \text{Yb}/1\% \text{Er}$ phosphor with the same crystallographic system of SMO is considered to be one of the most efficient UC/DS luminescent materials. To compare the emission intensity of $\text{YF}_3:20\% \text{Yb}/1\% \text{Er}$ and $\text{SMO}:20\% \text{Yb}/1\% \text{Er}$ phosphor, we have measured the temperature-dependent UC/DS spectra of $\text{YF}_3:20\% \text{Yb}/1\% \text{Er}$ and $\text{SMO}:20\% \text{Yb}/1\% \text{Er}$ under otherwise identical conditions (Supplementary Figures 6 and 7 newly added). It can be observed that the UC/DS intensity of $\text{YF}_3:20\% \text{Yb}/1\% \text{Er}$ phosphor decreased with increasing temperature, while the UC/DS intensity of $\text{SMO}:20\% \text{Yb}/1\% \text{Er}$ phosphor increased markedly with increasing the temperature. Specifically, the overall UC/DS intensity of $\text{YF}_3:20\% \text{Yb}/1\% \text{Er}$ is much higher than that of $\text{SMO}:20\% \text{Yb}/1\% \text{Er}$ at 298 K. Nevertheless, the overall UC and DS intensity of $\text{SMO}:20\% \text{Yb}/1\% \text{Er}$ phosphor are 4.5 and 12.9 times higher than that of $\text{YF}_3:20\% \text{Yb}/1\% \text{Er}$ counterpart at 773 K, respectively. **These results explicitly validate the superiority of the SMO:Yb/Er as novel luminescent materials over existing PTE phosphors, particularly at high temperatures.** The above discussions have been added in lines 1-13 of p. 10 and Supplementary Figures 6 and 7 newly added in the Supporting Information.

Supplementary Figure 6| Upconversion emission spectra of SMO:20%Yb³⁺/1%Er³⁺ phosphor and YF₃:20%Yb³⁺/1%Er³⁺ phosphor. a. XRD patterns of YF₃:20%Yb/1%Er phosphor. **b.** Upconversion emission spectra of the YF₃:20%Yb/1%Er as a function of temperature under 980 nm excitation. **c.** Comparison of the relative integrated intensity of the upconversion emission of the YF₃:20%Yb/1%Er and SMO:20%Yb/1%Er phosphor, wherein the data of SMO:20%Yb/1%Er originate from Figure 3b.

Supplementary Figure 7| Downshifting emission spectra of SMO:20%Yb³⁺/1%Er³⁺ phosphor and YF₃:20%Yb³⁺/1%Er³⁺ phosphor. a. and b. Downshifting emission spectra of the YF₃:20%Yb/1%Er and SMO:20%Yb/1%Er phosphor as a function of temperature under 980 nm excitation. **c.** Comparison of the relative integrated intensity of the downshifting emission spectra of the YF₃:20%Yb/1%Er and SMO:20%Yb/1%Er phosphor. The enlarged data of YF₃:20%Yb/1%Er is displayed on the right side.

Comment #12:

Lastly but not the least, this amazing sensitivity occurs at the exact same temperatures where the emission intensities are very low. Thus, at the points of the highest relative sensitivity, the sensing might be just possible, as the uncertainty in measurement will be great. This was discussed in the article: <https://onlinelibrary.wiley.com/doi/full/10.1002/adts.202000176>. This last point significantly diminishes the firstly claimed importance of the obtained results. I invite the authors to be honest in reporting the sensitivities only on the practically usable range. Also, taking everything in consideration, there is a must that the authors

report on the temperature resolutions at the whole temperature range, and unfortunately this means reporting on the uncertainties that should be evaluated by at least several temperatures. Uncertainties should be given as the standard deviation of at least 10 consecutive measurements at the same temperature. This temperature resolutions will show the true usability of the sensor, and my assumption is that the temperature resolution curve will be different than the sensitivity curves.

Response:

Many thanks for this valuable suggestion. Following the reviewer's suggestion, we measured the luminescence lifetimes of our phosphors at four temperature points (303, 313, 333, 363 K). All the PL decays for each temperature point were measured independently for ten times under identical conditions to yield the average value. Temperature resolution discussion via data analysis of the 10 consecutive measurements has been newly added at lines 1-14 in p. 16 of the revised manuscript. A slight change in sensitivity was obtained by re-fitting the data by adding data pointing. **From Figures 5a and 5b, it is obvious that the data become dense at the temperature range from 298 to 373 K, which can improve the data accuracy via decreasing the fitting error. The optimal S_a and S_r were determined to be as high as $53.0 \mu\text{sK}^{-1}$ and $12.3\% \text{K}^{-1}$, respectively.** The fitting results have been updated in the revised manuscript. The suggested reference (*Adv. Theory Simul.* 3, 2000176 (2020)) has been cited as ref. 45 for the discussion of temperature uncertainty.

Figure 5| Lifetime-based luminescence thermometry in 20%Yb³⁺/1%Er³⁺-codoped SMO. a. Experimentally measured and exponentially fitted plots of lifetime τ of SMO:20%Yb/1%Er at different temperatures. **b.** Calculated absolute sensitivity S_a and relative sensitivity S_r versus temperature based on the SMO:20% Yb/1%Er. **d.** Thermal dependence of temperature uncertainty for SMO:20%Yb/1%Er.

Moreover, the temperature uncertainty (δT) is an important parameter to assess the performance of a thermometer since it includes not only the relative sensitivity but also the error on the luminescence lifetime ($\delta\tau$)^{55,56}. δT is calculated as follows:

$$\delta T = \frac{1}{S_r} \cdot \frac{\delta\tau}{\tau} \quad (6)$$

Where $\delta\tau/\tau$ is the uncertainty in the calculation of τ (determined as a standard deviation in ten measurements of τ at the same temperature), S_r is the relative

sensitivity of luminescence thermometer. Temperature-dependent luminescence decay curves of SMO:20%Yb/1%Er phosphors by ten measurements at the same temperature are shown in the Supplementary Figure 15 newly added. Thermal dependence of temperature uncertainty for SMO:20%Yb/1%Er is presented in Figure 6d. The minimum value of δT is 0.11 K even at 623 K. **As such, although there is a relatively low value of S_r at high-temperature range, the obviously longer luminescence lifetime (τ) can effectively reduce the value of $\delta\tau/\tau$, resulting in lower δT .** It should be noted that it is possible to keep the temperature uncertainty below a threshold of 0.33 K throughout the whole studied temperature range (298-623 K). The δT threshold of SMO:20%Yb/1%Er for LLT is much lower than that (0.7 K) of cubic-phase LiLuF_4 :18%Yb³⁺/2% Er³⁺ nanocrystals for the conventional intensity-based thermometry. The δT threshold of SMO:20%Yb/1%Er is comparative to that (0.46) of LiLuF_4 :18%Yb³⁺/2% Er³⁺@SiO₂ (three shells) synthesized with the complicated condition. **All these results indicate that SMO:Yb/Er phosphor can be explored as a kind of ideal lifetime-based luminescence thermometry with high S_r and low δT , which validate the superiority and applicability of the proposed SMO:Yb/Er for luminescence thermometry.**

Draft Only

Supplementary Figure 15| Temperature-dependent luminescence decay curves of ${}^4I_{13/2}$ excited states of Er^{3+} in 20% Yb^{3+} /1% Er^{3+} -codoped SMO at different temperatures from 298 K to 623 K. Each temperature point was measured by ten consecutive measurements.

Reply to the Comments of Reviewer #3

General comment:

The authors present a two-dimensional negative thermal phosphor SMO:Yb/Er systematic study. They present a temperature dependent Raman, synchrotron X-ray diffraction and luminescence investigation. The thermally boosted UC and DS luminescence mechanism was investigated. The authors claim that the luminescence lifetime temperature dependency enables ratiometric thermometry with high relative sensitivity; an important topic in current research. This is a comprehensive work, very well written and with insights into the Ln³⁺-doped phosphors for potential applications.

Although they address an important topic some issues, that I refer below, need further clarification.

Response: We greatly appreciate the reviewer for his/her positive comments on our manuscript and the efforts to improve the quality of our manuscript. We have made all the requested changes in the revised manuscript according to the reviewer's comments. Particularly, the mechanism for the NTE behaviour of SMO:Yb³⁺/Er³⁺ has been clarified by means of more characterizations. Some inappropriate statement or unclear writing has been rephrased throughout the manuscript. A point-by-point response is noted below.

Comment #1:

Authors should comment, if known, what is the mechanism for the NTE behaviour in this system, whether it is structural, or electronically based for instance?

Response:

Many thanks for this valuable suggestion. Following the reviewer's suggestion, the structural rigid unit mode has been added to explain the mechanism for the NTE behavior of SMO:Yb³⁺/Er³⁺. The NTE behavior in this system is attributed to the structural variability with temperature. To shed more light on the structure of RE³⁺-doped SMO, rigid unit model for RE³⁺-doped SMO has been given as Figure 2e-2g (*Front. Phys.*, 16, 53302 (2021); *Rep. Prog. Phys.*, 79, 066503 (2016)). With the temperature increased from 420 to 623 K, the distortion of the REO₆ octahedron and twist of the RE-O-Mo may induce a variation in the rigid unit mode, resulting in a decrease ratio in the distance of RE-Mo along the *a* and *c* axes by 0.26% and 0.10%, as well as an increased ratio in the distance of RE-Mo along *b* axis by 0.02% (Figure 3d). **Note that the decreased amplitudes in the distance of RE-Mo along the *a* and *c* axes are one order larger than that along the *b* axis. Such an alteration causes not only shrinkage of lattice but also a local distortion of RE with increasing the temperature.** The angle of RE-Mo-RE nearly keeps unchanged at the temperature range of 420-623 K, which indicates that the structure of Sc₂Mo₃O₁₂ is rigid. The above discussions have been added in lines 17-24 of p. 5 in the revised manuscript.

Figure 3. **e.** Rigid unit mode model for $\text{Sc}_2\text{Mo}_3\text{O}_{12}:20\%\text{Yb}^{3+}/1\%\text{Er}^{3+}$ extracted from the unit cell structure. **f.** Temperature-dependence ratio δD of RE-Mo distances marked in the model ($\delta D = \frac{D_T - D_{420}}{D_{420}} \times 100\%$, D_T and D_{420} stand for RE-Mo distances of the given temperature and 420 K, respectively). **g.** Temperature-dependence ratio $\delta\theta$ of RE-Mo-RE angles marked in the model ($\delta\theta = \frac{\theta_T - \theta_{420}}{\theta_{420}} \times 100\%$, θ_T and θ_{420} stand for RE-Mo-RE angles of the given temperature and 420 K, respectively).

Comment #2:

What is supposed to drive the transition from PTE to NTE around 100 °C? The transition from PTE to NTE occurs at ca 100°C (figure 2c). How this transition impacts the luminescence properties and how this is compatible with the suggested thermally enhanced photoluminescence mechanism. It is important to clarify this issue. Some Raman modes appear to alter above 100 °C, can they be related with the NTE to PTE transition?

Response:

Many thanks for this valuable suggestion. **Such a transformation from PTE to NTE can be attributed to the existence of water molecules in the SMO:20%Yb/1%Er phosphors as revealed by the thermogravimetry (TG) analysis, *in situ* temperature-dependent Fourier transform infrared (FTIR) spectroscopy (Supplementary Figure 3 newly added), and temperature-dependent Raman spectra (Figure 2h).**

As shown in Supplementary Figure 3a, a weight loss of about 1% for $\text{SMO}:20\%\text{Yb}^{3+}/1\%\text{Er}^{3+}$ was detected by heating from 298 to 420 K. Generally, $\text{A}_2\text{Mo}_3\text{O}_{12}$ compounds with large A^{3+} cation size are highly hygroscopic and easily hydrated at ambient conditions. Therefore, $\text{Sc}_2\text{Mo}_3\text{O}_{12}$ compound with small Sc^{3+} (0.0745 nm, CN=6) is nonhygroscopic. For $\text{SMO}:20\%\text{Yb}^{3+}/1\%\text{Er}^{3+}$ phosphors with the substitution of Sc^{3+} by the big Yb^{3+} (0.0868 nm) and Er^{3+} (0.089 nm, CN=6), water molecules may be easily absorbed into the relatively large microchannels (Ref. 23, **Effect of water species on the phonon modes in orthorhombic $\text{Y}_2(\text{MoO}_4)_3$ revealed by Raman spectroscopy. *J. Phys. Chem. C*, 112, 6577- 6581 (2008).**). Accordingly, it

can be deduced that weight loss of about 1% for $\text{SMO:20\%Yb}^{3+}/1\%\text{Er}^{3+}$ may result from the removal of water molecules.

In the FTIR spectra, the typical absorbance peak at $3400\text{--}3500\text{ cm}^{-1}$, corresponding to the asymmetric vibration of -OH was markedly suppressed with the temperature from 298 to 420 K (Supplementary Figure 3b), which further verified the existence and removal of water molecules. As such, $\text{SMO:20\%Yb}^{3+}/1\%\text{Er}^{3+}$ phosphors with water molecules in the microchannels exhibit PTE behavior at low temperature (Ref.16, Negative thermal expansion: mechanisms and materials. *Front. Phys.*, 16, 53302 (2021)). With the temperature rising from 298 to 353 K, the water molecules remain in the microchannels. With an increase in temperature to around 353 K, water molecules begin to escape from the microchannels. As such, the $\text{SMO:20\%Yb}^{3+}/1\%\text{Er}^{3+}$ phosphors without water molecules in the microchannels exhibit intrinsic NTE character. When the temperature reaches 420 K, water molecules are completely removed and thus $\text{SMO:20\%Yb}^{3+}/1\%\text{Er}^{3+}$ phosphor may completely recover the NTE property.

Supplementary Figure 3| Water molecules analysis. a. Thermogravimetry curves of the $\text{SMO:20\%Yb}/1\%\text{Er}$ phosphor. **b.** Temperature-dependent Infrared spectra of $\text{SMO:20\%Yb}/1\%\text{Er}$ phosphor.

Supplementary Figure 4 c. Temperature-dependent changes of the unit cell volumes.

Figure 2 h. Temperature-dependent *in situ* Raman spectra within the temperature range from 298 to 773 K.

For the *in situ* temperature-dependent Raman spectra (Figure 2h), the Raman peaks of 330, 836, and 945 cm^{-1} are characteristic of the hydrated orthorhombic structure, indicative of water species residing in the microchannels of $\text{SMO:20\%Yb}^{3+}/1\%\text{Er}^{3+}$. As the temperature increases from 298 to 348 K, the intensity and position of these peaks remain unchanged, which reveals that the water molecules remain in the microchannels. As the temperature increases from 348 to 423 K, these peaks become weaker. Above 423 K, the peak at 945 cm^{-1} vanished, which reveals the complete removal of water molecules. These results are in agreement with the obtained results of the TG, FTIR spectra, as well as *in situ* temperature-dependent SXRD patterns (Supplementary Figures 3 and 4). Thus, it can be deduced that the temperature transition points of the onset and complete removal of water molecules from microchannels are ~ 348 and ~ 423 K, respectively. The above discussions have been added in lines 25-29 of p. 5 in the revised manuscript.

Based on these results, the photoluminescence mechanisms of SMO:Yb/Er phosphors at different temperatures have been updated. **For the temperature range of 298-353 K, the SMO:Yb/Er phosphors exhibit PTE due to the existence of water molecules in the microchannels. At higher temperatures above 353 K, the $\text{SMO:20\%Yb}^{3+}/1\%\text{Er}^{3+}$ phosphors exhibited thermally enhanced photoluminescence behavior due to the typical NTE properties of $\text{A}_2\text{Mo}_3\text{O}_{12}$ compounds** (Supplementary Figure 4c). Considering our topic is thermally boosted upconversion and downshifting luminescence in SMO:Yb/Er with two-dimensional negative thermal expansion, we have displayed the temperature range (353-773) of the thermally enhanced photoluminescence mechanism in the energy level diagram (Figure 5c). Please see also lines 2-5 of p. 12 in the revised manuscript and Supplementary Figure 3 newly added in the Supporting Information.

Figure 5| Thermally enhanced photoluminescence mechanism a. Temperature-dependent luminescence lifetime of ${}^2F_{5/2}$ excited state of Yb^{3+} in Yb^{3+}/Er^{3+} -codoped and Yb^{3+} -doped SMO, respectively. Temperature-dependent energy transfer efficiency of Yb^{3+} -to- Er^{3+} . **b.** Temperature-dependent lifetimes of ${}^2H_{11/2}$ (522 nm) and ${}^4I_{13/2}$ (1538 nm) excited states of Er^{3+} in SMO:20% Yb/1%Er, respectively. **c.** Energy level diagram of green UC and NIR DS emission showing the proposed temperature dependence of electronic transition and energy-transfer processes in SMO:Yb/Er with two-dimensional negative thermal expansion.

Comment #3:

From lines 182-183: "Moreover, the distance of $Yb^{3+}-Er^{3+}$ becomes shorter with an increase of the temperature. The ET processes between sensitizer (Yb^{3+}) to activator (Er^{3+}) are usually considered to occur through dipolar-dipolar interactions, whose ET efficiency is proportional to r^{-6} (r is the average donor-acceptor distance). The result indicates that the distances of $Yb^{3+}-Er^{3+}$ ions ($13.975 \approx$, at $25^\circ C$) decreased at higher temperature, which may benefit the improvement of the ET efficiency between $Yb^{3+}-Er^{3+}$."

And line 236 "Based on the above analysis, the thermally enhanced UC emission with the temperature from 25 to $200^\circ C$ is mainly governed by the ET from Yb^{3+} to Er^{3+} "

Authors should clarify the nature for thermally enhanced UC emission within the temperature range from 25 to $200^\circ C$, if this effect is mainly governed by the ET from Yb^{3+} to Er^{3+} , how it relates with the PTE of the system shown in Figure 2 observed between $25^\circ C$ and $100^\circ C$.

Response:

Many thanks for pointing out this problem. As indicated in the above analyses, SMO:20% Yb³⁺/1% Er³⁺ phosphors exhibit PTE behavior at the temperature below 353 K due to the existence of water molecules in the microchannels. When the temperature increases up to 353 K, water molecules begin to escape from the microchannels. As such, SMO:20% Yb³⁺/1% Er³⁺ phosphors exhibit thermally enhanced photoluminescence behavior due to the typical NTE properties of A₂Mo₃O₁₂ compounds. Please see also lines 22-29 of p. 4 in the revised manuscript and Supplementary Figure 3 newly added in the Supporting Information.

Comment #4:

From lines 46-49: " Most of such abnormal thermally enhanced UC luminescence is observed in Ln³⁺-doped inorganic materials with three-dimensional negative-thermal expansion characteristics (NTE)^{9, 10}, where all the three cell parameters of the doped crystals shrink at elevated temperature."

And below, lines 58-59 "with a unique two-dimensional NTE coefficient ($\alpha_a = -8.62 \times 10^{-6}/K$, $\alpha_b = 4.25 \times 10^{-6}/K$, $\alpha_c = -6.35 \times 10^{-6}/K$)"

In the context of this work, is there any effective advantage in two-dimensional negative-thermal expansion materials when compared with the three-dimensional ones? The authors could also mention recent works with uniaxial NTE.

Response:

Many thanks for this valuable suggestion. The advantage of two-dimensional (2D) negative-thermal expansion materials relative to the three-dimensional (3D) ones was not explicitly elucidated perhaps because of our unclear writing in the manuscript. For the previously reported RE³⁺-doped phosphors, the existence of 3D compression at elevated temperature can promote the energy transfer between sensitizers and activators. Meanwhile, such a 3D compression may also benefit the dissipation of the excitation energy in all directions of crystal sublattice to the lattice/surface defects, which deteriorates the luminescent emission of RE³⁺ ions. In this work, we propose a new class of NTE phosphors based on SMO:Yb/Er exhibiting a unique two-dimensional NTE coefficient. **With increasing the temperature, the SMO:Yb/Er phosphors experience an anisotropic two-dimensional shrinkage along the *a* and *c* axes at elevated temperature, whereas the distance between RE³⁺ ions increases along the *b* axis. As such, the detrimental energy migration of RE³⁺ dopants can be effectively confined within the two-dimensional layered structures (e.g., (020) lattice plane) at elevated temperatures.** Please see also lines 14-16 of p. 5 in the revised manuscript. Recent works with uniaxial NTE behavior have been cited as Refs. 12-15 (*Adv. Sci.* 2016, 3, 1600108; *Phys. Rev. B* 2020, 101, 104305; *Inorg. Chem.*, 56, 15101-15109 (2017); *Phys. Rev. B*, 96, 134113-134120 (2017)).

Comment #5:

Can the authors explain in what way the Energy migration between Ln³⁺ dopants is confined to two dimensions?

Response:

Thank you for pointing out this important issue. For RE³⁺-doped inorganic phosphors, the long-distance energy migration between RE³⁺ dopants depends on the distance between lattice positions occupied by RE³⁺ ions. As such, it is possible to minimize the depletion of excitation energy by employing a crystal lattice disfavoring the long-distance energy migration (*Nat. Mater.* **2014**, *13*, 157-162). In our work, the SMO:Yb/Er phosphors experience an anisotropic two-dimensional shrinkage along the *a* and *c* axes at elevated temperature, whereas the distance between RE³⁺ ions increases along with the *b* axis. Under such circumstances, the migration of excitation energy between RE³⁺ dopants may be effectively minimized long the *b* axis. Thus, more energy migration between RE³⁺ dopants is confined to the two-dimensional layered structures (*e.g.*, (020) lattice plane) at elevated temperatures. Please see also lines 21-23 of p. 2 in the revised manuscript.

Comment #6:

From lines 231-233. The promoted A_r and inhibited A_{nr} in the Ln³⁺ dopant indicate that lattice distortion enhances a crystal field with odd parity and modifies local symmetry of activator ions via the NTE effect.

The authors mention the existence of lattice distortion and a change in the local symmetry of activator ions via the NTE effect. Can these local distortions be substantiated via the temperature-dependent Raman data?

Response:

Many thanks for this valuable suggestion. Raman spectroscopy is a valuable tool to study the phonon modes of NTE materials and evaluate the change of local structure. ***In situ* temperature-dependent Raman spectra are shown in Figure 2h.** The Raman peaks of 330, 836, and 945 cm⁻¹ are characteristic of the hydrated orthorhombic structure, indicative of water species residing in the microchannels of SMO:20% Yb³⁺/1% Er³⁺. As the temperature increases from 298 to 348 K, the intensity and position of these peaks nearly remain unchanged, which reveals that the water molecules were not removed from the microchannels. As the temperature increases from 348 to 423 K, these peaks become weaker. Above 423 K, the peak at 945 cm⁻¹ vanished, which demonstrated that water molecules are completely removed. These results are in agreement with the obtained results of the TG, FTIR spectra, as well as *in situ* temperature-dependent SXRD patterns (Supplementary Figures 3 and 4). Thus, it can be deduced that the temperature transition points of the onset and complete removal of water molecules from microchannels are ~348 and ~423 K, respectively. In addition, as the temperature increases, a Raman peak with a frequency of 341 cm⁻¹

(ν_4) exhibits a blue shift (Figure 2h), suggesting that this mode is the origin of NTE in SMO:Yb/Er. The median frequency at 510 cm^{-1} and 579 cm^{-1} is induced by the disorder of MoO_4 tetrahedra because of the incorporation of the Yb^{3+} and Er^{3+} into the lattice, which will affect the local structure of the doped Yb^{3+} and Er^{3+} ions. Note that the temperature coefficients of high phonon frequencies ($511, 579, 813, 979\text{ cm}^{-1}$) are negative above 348 K (Figures 2i and 2j), verifying strong anharmonic stretching/bending of MoO_4 tetrahedra. These results indicate that high phonon frequencies contribute also to the NTE behavior of the sample. Moreover, the reduction of maximum phonon energy (979 cm^{-1}) may benefit the luminescence of RE^{3+} at elevated temperatures due to the suppressed nonradiative relaxation. Please see also lines 1-17 of p. 7 in the revised manuscript.

Comment #7

Can this effect be witnessed by the other techniques that would perceive the negative thermal expansion at the local scale? The authors could comment on that.

Response:

The negative thermal expansion at the local scale has also been witnessed by the other techniques like atomically resolved scanning tunnel microscope (STM) topographic image. We have cited the corresponding reference about the characterization of negative thermal expansion as Ref.13 (*Phys. Rev. B*, 101, 104305 (2020)) in the revised manuscript.

Comment #8:

Along the manuscript:

What is the degree of Sc substitution in the samples that were measured in the presented results Figure 2,3,4,5? This is only specified in the labels of the Figures presented in the Supplementary material.

Response:

Many thanks for this valuable suggestion. We have provided the degree of Sc substitution (Sc substituted by 20% Yb and 1%Er) in the samples for Figures 2-5 in the revised manuscript.

Comment #9:

line 18 "decrease at the higher temperature" author should substitute "decrease for high temperatures"

Response:

"decrease at the higher temperature" has been revised as "decrease for high

temperatures" in the revised manuscript. Please see also line 20 of p. 1 in the revised manuscript.

Comment #10:

line 58 and 103 Where is the abbreviation "NET" defined? Do the authors mean "NTE"?

Response:

Thank you for pointing out this typo. The abbreviation "NET" has been revised as "NTE" in the revised manuscript.

Comment #11:

Line 121-122: What is the value of Xc used for the calculations?

Response:

According to the second reviewer's suggestion, the ionic distance was updated, which was determined by the crystallographic information file (CIF) of *Rietveld* refinement on the basis of the SXR D patterns. Therefore, Equation 1 used for the calculation has been deleted.

Comment #12:

Line 133 -135 "The observed mode with a frequency of 341 cm^{-1} corresponds to the bending mode of MoO_4 tetrahedra. When the temperature increases, this Raman peak exhibits a blue shift (Figure 2f). The median frequency B1g mode of SMO is observed at 511 cm^{-1} " Is this a typo? Figure 2f is not from the 341 cm^{-1} mode. The sentence should be improved.

Response:

Thank you for pointing out this typo. "this Raman peak exhibits a blue shift (Figure 2f)" has been revised as "this Raman peak exhibits a blue shift (Figure 2h)". Please see also line 10 of p. 7 in the revised manuscript.

Comment #13:

The authors refer the studied systems as "the structures of the SMO:Yb/Er samples with different dopant concentrations", doping generally refers to a small concentrations, here the degree of ionic Sc substitution by Yb and Er cations reaches the high value of 25% or 8%. The term chemical substitution would be more appropriated.

Response:

Thanks. "the structures of the SMO:Yb/Er samples with different dopant concentrations" has been revised as "the structures of the SMO:Yb/Er samples with different substituted concentrations".

Draft Only

REVIEWERS' COMMENTS

Reviewer #1 (Remarks to the Author):

The authors responded correctly to my comments. The work does not contain any factual errors. In my opinion, it can be accepted for publication in its current form

Reviewer #2 (Remarks to the Author):

The authors have adequately addressed all of the reviewer's comments, thus I recommend this article for publication.

Reviewer #3 (Remarks to the Author):

The paper is much improved, and it addresses the issues that concerned me. In my opinion is ready for publication.